# TRAINING DEEP NEURAL NETWORKS WITH PARTIALLY ADAPTIVE MOMENTUM

## ABSTRACT

Adaptive gradient methods, which adopt historical gradient information to automatically adjust the learning rate, despite the nice property of fast convergence, have been observed to generalize worse than stochastic gradient descent (SGD) with momentum in training deep neural networks. This leaves how to close the generalization gap of adaptive gradient methods an open problem. In this work, we show that adaptive gradient methods such as Adam, Amsgrad, are sometimes "over adapted". We design a new algorithm, called *Partially adaptive momentum estimation method*, which unifies the Adam/Amsgrad with SGD by introducing a partial adaptive parameter $p$, to achieve the best from both worlds. We also prove the convergence rate of our proposed algorithm to a stationary point in the stochastic nonconvex optimization setting. Experiments on standard benchmarks show that our proposed algorithm can maintain fast convergence rate as Adam/Amsgrad while generalizing as well as SGD in training deep neural networks. These results would suggest practitioners pick up adaptive gradient methods once again for faster training of deep neural networks.

## 1  INTRODUCTION

Stochastic gradient descent (SGD) is now one of the most dominant approaches for training deep neural networks (Goodfellow et al., 2016). In each iteration, SGD only performs one parameter update on a mini-batch of training examples. SGD is simple and has been proved to be efficient, especially for tasks on large datasets. In recent years, adaptive variants of SGD have emerged and shown their successes for their convenient automatic learning rate adjustment mechanism. Adagrad (Duchi et al., 2011) is probably the first along this line of research, and significantly outperforms vanilla SGD in the sparse gradient scenario. Despite the first success, Adagrad was later found to demonstrate degraded performance especially in cases where the loss function is nonconvex or the gradient is dense. Many variants of Adagrad, such as RMSprop (Hinton et al., 2012), Adam (Kingma & Ba, 2015), Adadelta (Zeiler, 2012), Nadam (Dozat, 2016), were then proposed to address these challenges by adopting exponential moving average rather than the arithmetic average used in Adagrad. This change largely mitigates the rapid decay of learning rate in Adagrad and hence makes this family of algorithms, especially Adam, particularly popular on various tasks. Recently, it has also been observed (Reddi et al., 2018) that Adam does not converge in some settings where rarely encountered large gradient information quickly dies out due to the "short momery" problem of exponential moving average. To address this issue, Amsgrad (Reddi et al., 2018) has been proposed to keep an extra "long term memory" variable to preserve the past gradient information and to correct the potential convergence issue in Adam. There are also some other variants of adaptive gradient method such as SC-Adagrad / SC-RMSprop (Mukkamala & Hein, 2017), which derives logarithmic regret bounds for strongly convex functions.

On the other hand, people recently found that for largely over-parameterized neural networks, e.g., more complex modern convolutional neural network (CNN) architectures such as VGGNet (He et al., 2016), ResNet (He et al., 2016), Wide ResNet (Zagoruyko & Komodakis, 2016), DenseNet (Huang et al., 2017), training with Adam or its variants typically generalizes worse than SGD with momentum, even when the training performance is better. In particular, people found that carefully-tuned SGD, combined with proper momentum, weight decay and appropriate learning rate decay schedules, can significantly outperform adaptive gradient algorithms eventually (Wilson et al., 2017). As a result, even though adaptive gradient methods are relatively easy to tune and converge

faster at the early stage, recent advances in designing neural network structures are all reporting their performances by training their models with SGD-momentum (He et al., 2016; Zagoruyko & Komodakis, 2016; Huang et al., 2017; Simonyan & Zisserman, 2014; Ren et al., 2015; Xie et al., 2017; Howard et al., 2017). Different from SGD, which adopts a universal learning rate for all coordinates, the effective learning rate of adaptive gradient methods, i.e., the universal base learning rate divided by the second order moment term, is different for different coordinates. Due to the normalization of the second order moment, some coordinates will have very large effective learning rates. To alleviate this problem, one usually chooses a smaller base learning rate for adaptive gradient methods than SGD with momentum. This makes the learning rate decay schedule less effective when applied to adaptive gradient methods, since a much smaller base learning rate will soon diminish after several rounds of decay. We refer to the above phenomenon as the "small learning rate dilemma" (see more details in Section 3).

With all these observations, a natural question is:

*Can we take the best from both Adam and SGD-momentum, i.e., design an algorithm that not only enjoys the fast convergence rate as Adam, but also generalizes as well as SGD-momentum?*

In this paper, we answer this question affirmatively. We close the generalization gap of adaptive gradient methods by proposing a new algorithm, called **p**artially **ada**ptive **m**omentum estimation (Padam) method, which unifies Adam/Amsgrad with SGD-momentum to achieve the best of both worlds, by a partially adaptive parameter. The intuition behind our algorithm is: by controlling the degree of adaptiveness, the base learning rate in Padam does not need to be as small as other adaptive gradient methods. Therefore, it can maintain a larger learning rate while preventing the gradient explosion. We note that there exist several studies (Zaheer et al., 2018; Loshchilov & Hutter, 2019; Luo et al., 2019) that also attempted to address the same research question. In detail, Yogi (Zaheer et al., 2018) studied the effect of adaptive denominator constant $\epsilon$ and minibatch size in the convergence of adaptive gradient methods. AdamW (Loshchilov & Hutter, 2019) proposed to fix the weight decay regularization in Adam by decoupling the weight decay from the gradient update and this improves the generalization performance of Adam. AdaBound (Luo et al., 2019) applies dynamic bound of learning rate on Adam and make them smoothly converge to a constant final step size as in SGD. Our algorithm is very different from Yogi, AdamW and AdaBound. Padam is built upon a simple modification of Adam without extra complicated algorithmic design and it comes with a rigorous convergence guarantee in the nonconvex stochastic optimization setting.

We highlight the main contributions of our work as follows:

• We propose a novel and simple algorithm Padam with a partially adaptive parameter, which resolves the "small learning rate dilemma" for adaptive gradient methods and allows for faster convergence, hence closing the gap of generalization.

• We provide a convergence guarantee for Padam in nonconvex optimization. Specifically, we prove that the convergence rate of Padam to a stationary point for stochastic nonconvex optimization is

$$O\left(\frac{\left(\sum_{i=1}^{d} \|\mathbf{g}_{1:T,i}\|_2\right)^{1/2}}{T^{3/4}} + \frac{d}{T}\right), \tag{1.1}$$

where $\mathbf{g}_1, \ldots, \mathbf{g}_T$ are the stochastic gradients and $\mathbf{g}_{1:T,i} = [g_{1,i}, g_{2,i}, \ldots, g_{T,i}]^\top$. When the stochastic gradients are $\ell_\infty$-bounded, (1.1) matches the convergence rate of SGD in terms of the rate of $T$.

• We also provide thorough experiments about our proposed Padam method on training modern deep neural architectures. We empirically show that Padam achieves the fastest convergence speed while generalizing as well as SGD with momentum. These results suggest that practitioners should pick up adaptive gradient methods once again for faster training of deep neural networks.

• Last but not least, compared with the recent work on adaptive gradient methods, such as Yogi (Zaheer et al., 2018), AdamW (Loshchilov & Hutter, 2019), AdaBound (Luo et al., 2019), our proposed Padam achieves better generalization performance than these methods in our experiments.

### 1.1 ADDITIONAL RELATED WORK

Here we review additional related work that is not covered before. Zhang et al. (2017) proposed a normalized direction-preserving Adam (ND-Adam), which changes the adaptive terms from individual dimensions to the whole gradient vector. Keskar & Socher (2017) proposed to improve

the generalization performance by switching from Adam to SGD. On the other hand, despite the great successes of adaptive gradient methods for training deep neural networks, the convergence guarantees for these algorithms are still understudied. Most convergence analyses of adaptive gradient methods are restricted to online convex optimization (Duchi et al., 2011; Kingma & Ba, 2015; Mukkamala & Hein, 2017; Reddi et al., 2018). A few recent attempts have been made to analyze adaptive gradient methods for stochastic nonconvex optimization. More specifically, Basu et al. (2018) proved the convergence rate of RMSProp and Adam when using deterministic gradient rather than stochastic gradient. Li & Orabona (2018) analyzed convergence rate of AdaGrad under both convex and nonconvex settings but did not consider more complicated Adam-type algorithms. Ward et al. (2018) also proved the convergence rate of AdaGrad under both convex and nonconvex settings without considering the effect of stochastic momentum. Chen et al. (2018) provided a convergence analysis for a class of Adam-type algorithms for nonconvex optimization. Zou & Shen (2018) analyzed the convergence rate of AdaHB and AdaNAG, two modified version of AdaGrad with the use of momentum. However, none of these results are directly applicable to Padam. Our convergence analysis in Section 4 is quite general and implies the convergence rate of Adam/AMSGrad for nonconvex optimization. In terms of learning rate decay schedule, Wu et al. (2018) studied the learning rate schedule via short-horizon bias. Xu et al. (2016); Davis et al. (2019) analyzed the convergence of stochastic algorithms with geometric learning rate decay. Ge et al. (2019) studied the learning rate schedule for quadratic functions.

**Notation:** Scalars are denoted by lower case letters, vectors by lower case bold face letters, and matrices by upper case bold face letters. For a vector $\mathbf{x} \in \mathbb{R}^d$, we denote the $\ell_2$ norm of $\mathbf{x}$ by $\|\mathbf{x}\|_2 = \sqrt{\sum_{i=1}^d x_i^2}$, the $\ell_\infty$ norm of $\mathbf{x}$ by $\|\mathbf{x}\|_\infty = \max_{i=1}^d |x_i|$. For a sequence of vectors $\{\mathbf{x}_j\}_{j=1}^t$, we denote by $x_{j,i}$ the $i$-th element in $\mathbf{x}_j$. We also denote $\mathbf{x}_{1:t,i} = [x_{1,i}, \ldots, x_{t,i}]^\top$. With slightly abuse of notation, for two vectors $\mathbf{a}$ and $\mathbf{b}$, we denote $\mathbf{a}^2$ as the element-wise square, $\mathbf{a}^p$ as the element-wise power operation, $\mathbf{a}/\mathbf{b}$ as the element-wise division and $\max(\mathbf{a}, \mathbf{b})$ as the element-wise maximum. We denote by $\text{diag}(\mathbf{a})$ a diagonal matrix with diagonal entries $a_1, \ldots, a_d$. Given two sequences $\{a_n\}$ and $\{b_n\}$, we write $a_n = O(b_n)$ if there exists a positive constant $C$ such that $a_n \leq Cb_n$ and $a_n = o(b_n)$ if $a_n/b_n \to 0$ as $n \to \infty$. Notation $\widetilde{O}(\cdot)$ hides logarithmic factors.

## 2 REVIEW OF ADAPTIVE GRADIENT METHODS

Various adaptive gradient methods have been proposed in order to achieve better performance on various stochastic optimization tasks. Adagrad (Duchi et al., 2011) is among the first methods with adaptive learning rate for each individual dimension, which motivates the research on adaptive gradient methods in the machine learning community. In detail, Adagrad[1] adopts the following update form:

$$\mathbf{x}_{t+1} = \mathbf{x}_t - \alpha_t \frac{\mathbf{g}_t}{\sqrt{\mathbf{v}_t}}, \quad \text{where } \mathbf{v}_t = \frac{1}{t}\sum_{j=1}^t \mathbf{g}_j^2, \qquad \text{(Adagrad)}$$

where $\mathbf{g}_t$ stands for the stochastic gradient $\nabla f_t(\mathbf{x}_t)$, and $\alpha_t = \alpha/\sqrt{t}$ is the step size. In this paper, we call $\alpha_t$ *base learning rate*, which is the same for all coordinates of $\mathbf{x}_t$, and we call $\alpha_t/\sqrt{v_{t,i}}$ *effective learning rate* for the $i$-th coordinate of $\mathbf{x}_t$, which varies across the coordinates. Adagrad is proved to enjoy a huge gain in terms of convergence especially in sparse gradient situations. Empirical studies also show a performance gain even for non-sparse gradient settings. RMSprop (Hinton et al., 2012) follows the idea of adaptive learning rate and it changes the arithmetic averages used for $\mathbf{v}_t$ in Adagrad to exponential moving averages. Even though RMSprop is an empirical method with no theoretical guarantee, the outstanding empirical performance of RMSprop raised people's interests in exponential moving average variants of Adagrad. Adam (Kingma & Ba, 2015)[2] is the most popular exponential moving average variant of Adagrad. It combines the idea of RMSprop and momentum acceleration, and takes the following update form:

$$\mathbf{x}_{t+1} = \mathbf{x}_t - \alpha_t \frac{\mathbf{m}_t}{\sqrt{\mathbf{v}_t}} \quad \text{where } \mathbf{m}_t = \beta_1 \mathbf{m}_{t-1} + (1 - \beta_1)\mathbf{g}_t, \mathbf{v}_t = \beta_2 \mathbf{v}_{t-1} + (1 - \beta_2)\mathbf{g}_t^2. \quad \text{(Adam)}$$

---

[1]The formula is equivalent to the one from the original paper (Duchi et al., 2011) after simple manipulations.
[2]Here for simplicity and consistency, we ignore the bias correction step in the original paper of Adam. Yet adding the bias correction step will not affect the argument in the paper.

Adam also requires $\alpha_t = \alpha/\sqrt{t}$ for the sake of convergence analysis. In practice, any decaying step size or even constant step size works well for Adam. Note that if we choose $\beta_1 = 0$, Adam basically reduces to RMSprop. Reddi et al. (2018) identified a non-convergence issue in Adam. Specifically, Adam does not collect long-term memory of past gradients and therefore the effective learning rate could be increasing in some cases. They proposed a modified algorithm namely Amsgrad. More specifically, Amsgrad adopts an additional step to ensure the decay of the effective learning rate $\alpha_t/\sqrt{\widehat{\mathbf{v}}_t}$, and its key update formula is as follows:

$$\mathbf{x}_{t+1} = \mathbf{x}_t - \alpha_t \frac{\mathbf{m}_t}{\sqrt{\widehat{\mathbf{v}}_t}}, \text{ where } \widehat{\mathbf{v}}_t = \max(\widehat{\mathbf{v}}_{t-1}, \mathbf{v}_t), \qquad \text{(Amsgrad)}$$

$\mathbf{m}_t$ and $\mathbf{v}_t$ are the same as Adam. By introducing the $\widehat{\mathbf{v}}_t$ term, Reddi et al. (2018) corrected some mistakes in the original proof of Adam and proved an $O(1/\sqrt{T})$ convergence rate of Amsgrad for convex optimization. Note that all the theoretical guarantees on adaptive gradient methods (Adagrad, Adam, Amsgrad) are only proved for convex functions.

## 3 THE PROPOSED ALGORITHM

In this section, we propose a new algorithm for bridging the generalization gap for adaptive gradient methods. Specifically, we introduce a partial adaptive parameter $p$ to control the level of adaptiveness of the optimization procedure. The proposed algorithm is displayed in Algorithm 1.

---

**Algorithm 1** Partially adaptive momentum estimation method (Padam)

> **input:** initial point $\mathbf{x}_1 \in \mathcal{X}$; step sizes $\{\alpha_t\}$; adaptive parameters $\beta_1, \beta_2, p \in (0, 1/2]$
> set $\mathbf{m}_0 = \mathbf{0}, \mathbf{v}_0 = \mathbf{0}, \widehat{\mathbf{v}}_0 = \mathbf{0}$
> **for** $t = 1, \ldots, T$ **do**
>     $\mathbf{g}_t = \nabla f(\mathbf{x}_t, \xi_t)$
>     $\mathbf{m}_t = \beta_1 \mathbf{m}_{t-1} + (1 - \beta_1)\mathbf{g}_t$
>     $\mathbf{v}_t = \beta_2 \mathbf{v}_{t-1} + (1 - \beta_2)\mathbf{g}_t^2$
>     $\widehat{\mathbf{v}}_t = \max(\widehat{\mathbf{v}}_{t-1}, \mathbf{v}_t)$
>     $\mathbf{x}_{t+1} = \mathbf{x}_t - \alpha_t \cdot \mathbf{m}_t / \widehat{\mathbf{v}}_t^p$
> **end for**
> **Output:** Choose $\mathbf{x}_{\text{out}}$ from $\{\mathbf{x}_t\}, 2 \leq t \leq T$ with probability $\alpha_{t-1}/\left(\sum_{i=1}^{T-1} \alpha_i\right)$

---

In Algorithm 1, $\mathbf{g}_t$ denotes the stochastic gradient and $\widehat{\mathbf{v}}_t$ can be seen as a moving average over the second order moment of the stochastic gradients. As we can see from Algorithm 1, the key difference between Padam and Amsgrad (Reddi et al., 2018) is that: while $\mathbf{m}_t$ is still the momentum as in Adam/Amsgrad, it is now "partially adapted" by the second order moment, i.e.,

$$\mathbf{x}_{t+1} = \mathbf{x}_t - \alpha_t \frac{\mathbf{m}_t}{\widehat{\mathbf{v}}_t^p}, \text{ where } \widehat{\mathbf{v}}_t = \max(\widehat{\mathbf{v}}_{t-1}, \mathbf{v}_t). \qquad \text{(Padam)}$$

We call $p \in [0, 1/2]$ the partially adaptive parameter. Note that $1/2$ is the largest possible value for $p$ and a larger $p$ will result in non-convergence in the proof (see the proof details in the supplementary materials). When $p \to 0$, Algorithm 1 reduces to SGD with momentum[3] and when $p = 1/2$, Algorithm 1 is exactly Amsgrad. Therefore, Padam indeed unifies Amsgrad and SGD with momentum.

With the notations defined above, we are able to formally explain the "small learning rate dilemma". In order to make things clear, we first emphasize the relationship between adaptiveness and learning rate decay. We refer the actual learning rate applied to $\mathbf{m}_t$ as the effective learning rate, i.e., $\alpha_t/\widehat{\mathbf{v}}_t^p$. Now suppose that a learning rate decay schedule is applied to $\alpha_t$. If $p$ is large, then at early stages, the effective learning rate $\alpha_t/\widehat{v}_{t,i}^p$ could be fairly large for certain coordinates with small $\widehat{v}_{t,i}$ value[4]. To prevent those coordinates from overshooting we need to enforce a smaller $\alpha_t$, and therefore the base learning rate must be set small (Keskar & Socher, 2017; Wilson et al., 2017). As a result, after several rounds of decaying, the learning rates of the adaptive gradient methods are too small

---

[3]The only difference between Padam with $p = 0$ and SGD-Momentum is an extra constant factor $(1 - \beta_1)$, which can be moved into the learning rate such that the update rules for these two algorithms are identical.

[4]The coordinate $\widehat{v}_{t,i}$'s are much less than 1 for most commonly used network architectures. See Figure 5 in the supplementary materials.

to make any significant progress in the training process[5]. We call this phenomenon "small learning rate dilemma". It is also easy to see that the larger $p$ is, the more severe "small learning rate dilemma" is. This suggests that intuitively, we should consider using Padam with a proper adaptive parameter $p$, and choosing $p < 1/2$ can potentially make Padam suffer less from the "small learning rate dilemma" than Amsgrad, which justifies the range of $p$ in Algorithm 1. We will show in our experiments (Section 5) that Padam with $p < 1/2$ can adopt an equally large base learning rate as SGD with momentum.

Note that even though in Algorithm 1, the choice of $\alpha_t$ covers different choices of learning rate decay schedule, the main focus of this paper is not about finding the best learning rate decay schedule, but designing a new algorithm to control the adaptiveness for better empirical generalization result. In other words, our focus is not on $\alpha_t$, but on $\widehat{\mathbf{v}}_t$. For this reason, we simply fix the learning rate decay schedule for all methods in the experiments to provide a fair comparison for different methods.

Figure 1 shows the comparison of test error performances under the different partial adaptive parameter $p$ for ResNet on both CIFAR-10 and CIFAR-100 datasets. We can observe that a larger $p$ will lead to fast convergence at early stages and worse generalization performance later, while a smaller $p$ behaves more like SGD with momentum: slow in early stages but finally catch up. With a proper choice of $p$ (e.g., $1/8$ in this case), Padam can

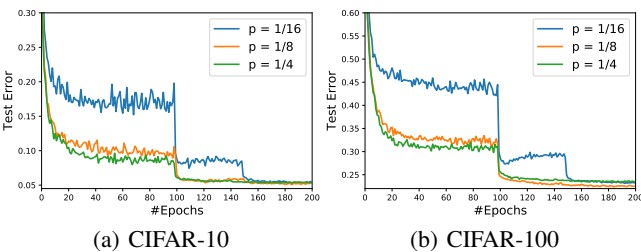

(a) CIFAR-10        (b) CIFAR-100

Figure 1: Performance comparison of Padam with different choices of $p$ for training ResNet on CIFAR-10 / CIFAR-100.

obtain the best of both worlds. Note that besides Algorithm 1, our partially adaptive idea can also be applied to other adaptive gradient methods such as Adagrad, Adadelta, RMSprop, AdaMax (Kingma & Ba, 2015). For the sake of conciseness, we do not list the partially adaptive versions for other adaptive gradient methods here. We also would like to comment that Padam is totally different from the $p$-norm generalized version of Adam in Kingma & Ba (2015), which induces AdaMax method when $p \to \infty$. In their case, $p$-norm is used to generalize 2-norm of their current and past gradients while keeping the scale of adaptation unchanged. In sharp contrast, we intentionally change (reduce) the scale of the adaptive term in Padam to get better generalization performance. Finally, note that in Algorithm 1 we remove the bias correction step used in the original Adam paper following Reddi et al. (2018). Nevertheless, our arguments and theory are applicable to the bias correction version as well.

## 4   CONVERGENCE ANALYSIS OF THE PROPOSED ALGORITHM

In this section, we establish the convergence theory of Algorithm 1 in the stochastic nonconvex optimization setting, i.e., we aim at solving the following stochastic nonconvex optimization problem

$$\min_{\mathbf{x} \in \mathbb{R}^d} f(\mathbf{x}) := \mathbb{E}_\xi \big[ f(\mathbf{x}; \xi) \big],$$

where $\xi$ is a random variable satisfying certain distribution, $f(\mathbf{x}; \xi) : \mathbb{R}^d \to \mathbb{R}$ is a $L$-smooth nonconvex function. In the stochastic setting, one cannot directly access the full gradient of $f(\mathbf{x})$. Instead, one can only get unbiased estimators of the gradient of $f(\mathbf{x})$, which is $\nabla f(\mathbf{x}; \xi)$. This setting has been studied in Ghadimi & Lan (2013; 2016). We first introduce the following assumptions.

**Assumption 4.1** (Bounded Gradient). $f(\mathbf{x}) = \mathbb{E}_\xi f(\mathbf{x}; \xi)$ has $G_\infty$-bounded stochastic gradient. That is, for any $\xi$, we assume that $\|\nabla f(\mathbf{x}; \xi)\|_\infty \leq G_\infty$.

It is worth mentioning that Assumption 4.1 is slightly weaker than the $\ell_2$-boundedness assumption $\|\nabla f(\mathbf{x}; \xi)\|_2 \leq G_2$ used in Reddi et al. (2016); Chen et al. (2018). Since $\|\nabla f(\mathbf{x}; \xi)\|_\infty \leq \|\nabla f(\mathbf{x}; \xi)\|_2 \leq \sqrt{d}\|\nabla f(\mathbf{x}; \xi)\|_\infty$, the $\ell_2$-boundedness assumption implies Assumption 4.1 with

---

[5]This does not mean the learning rate decay schedule weakens adaptive gradient method. On the contrary, applying the learning rate decay schedule still gives performance boost to the adaptive gradient methods in general but this performance boost is not as significant as SGD + momentum.

$G_\infty = G_2$. Meanwhile, $G_\infty$ will be tighter than $G_2$ by a factor of $\sqrt{d}$ when each coordinate of $\nabla f(\mathbf{x}; \xi)$ almost equals to each other.

**Assumption 4.2** (*L*-smooth). $f(\mathbf{x}) = \mathbb{E}_\xi f(\mathbf{x}; \xi)$ is *L*-smooth: for any $\mathbf{x}, \mathbf{y} \in \mathbb{R}^d$, it satisfied that $\left| f(\mathbf{x}) - f(\mathbf{y}) - \langle \nabla f(\mathbf{y}), \mathbf{x} - \mathbf{y} \rangle \right| \leq \frac{L}{2} \|\mathbf{x} - \mathbf{y}\|_2^2$.

Assumption 4.2 is frequently used in analysis of gradient-based algorithms. It is equivalent to the *L*-gradient Lipschitz condition, which is often written as $\|\nabla f(\mathbf{x}) - \nabla f(\mathbf{y})\|_2 \leq L\|\mathbf{x} - \mathbf{y}\|_2$. Next we provide the main convergence rate result for our proposed algorithm.

**Theorem 4.3** (Padam). In Algorithm 1, suppose that $p \in [0, 1/2]$, $\beta_1 < \beta_2^{2p}$ and $\alpha_t = \alpha$ for $t = 1, \dots, T$, under Assumptions 4.1 and 4.2, let $\Delta f = f(\mathbf{x}_1) - \inf_{\mathbf{x}} f(\mathbf{x})$, for any $q \in [\max\{0, 4p - 1\}, 1]$, the output $\mathbf{x}_{\text{out}}$ of Algorithm 1 satisfies that

$$\mathbb{E}\left[\|\nabla f(\mathbf{x}_{\text{out}})\|_2^2\right] \leq \frac{M_1}{T\alpha} + \frac{M_2 d}{T} + \frac{M_3 \alpha d^q}{T^{(1-q)/2}} \mathbb{E}\left(\sum_{i=1}^d \|\mathbf{g}_{1:T,i}\|_2\right)^{1-q}, \tag{4.1}$$

where

$$M_1 = 2G_\infty^{2p}\Delta f, \; M_2 = \frac{4G_\infty^{2+2p}\mathbb{E}\|\widehat{\mathbf{v}}_1^{-p}\|_1}{d(1-\beta_1)} + 4G_\infty^2, \; M_3 = \frac{4LG_\infty^{1+q-2p}}{(1-\beta_2)^{2p}} + \frac{8LG_\infty^{1+q-2p}(1-\beta_1)}{(1-\beta_2)^{2p}(1-\beta_1/\beta_2^{2p})}\left(\frac{\beta_1}{1-\beta_1}\right)^2.$$

**Remark 4.4.** From Theorem 4.3, we can see that $M_1$ and $M_3$ are independent of the number of iterations $T$ and dimension $d$. In addition, if $\|\widehat{\mathbf{v}}_1^{-1}\|_\infty = O(1)$, it is easy to see that $M_2$ also has an upper bound that is independent of $T$ and $d$.

The following corollary is a special case of Theorem 4.3 when $p \in [0, 1/4]$ and $q = 0$.

**Corollary 4.5.** Under the same conditions in Theorem 4.3, if $p \in [0, 1/4]$, Padam's output satisfies

$$\mathbb{E}\left[\|\nabla f(\mathbf{x}_{\text{out}})\|_2^2\right] \leq \frac{M_1}{T\alpha} + \frac{M_2 d}{T} + \frac{M_3'\alpha}{\sqrt{T}} \mathbb{E}\left(\sum_{i=1}^d \|\mathbf{g}_{1:T,i}\|_2\right), \tag{4.2}$$

where $M_1$ and $M_2$ and $\Delta f$ are the same as in Theorem 4.3, and $M_3'$ is defined as follows:

$$M_3' = \frac{4LG_\infty^{1-2p}}{(1-\beta_2)^{2p}} + \frac{8LG_\infty^{1-2p}(1-\beta_1)}{(1-\beta_2)^{2p}(1-\beta_1/\beta_2^{2p})}\left(\frac{\beta_1}{1-\beta_1}\right)^2.$$

**Remark 4.6.** Corollary 4.5 simplifies the result of Theorem 4.3 by choosing $q = 0$ under the condition $p \in [0, 1/4]$. We remark that this choice of $q$ is optimal in an important special case studied in Duchi et al. (2011); Reddi et al. (2018): when the gradient vectors are sparse, we assume that $\sum_{i=1}^d \|\mathbf{g}_{1:T,i}\|_2 \ll \sqrt{dT}$. Then for $q > 0$, it follows that

$$\frac{\sum_{i=1}^d \|\mathbf{g}_{1:T,i}\|_2}{T} \ll \frac{d^q\left(\sum_{i=1}^d \|\mathbf{g}_{1:T,i}\|_2\right)^{1-q}}{T^{1-q/2}}. \tag{4.3}$$

(4.3) implies that the upper bound provided by (4.2) is strictly better than (4.1) with $q > 0$. Therefore when the gradient vectors are sparse, Padam achieves faster convergence when $p \in [0, 1/4]$.

**Remark 4.7.** We show the convergence rate under different choices of step size $\alpha$. If

$$\alpha = \Theta\left(T^{1/4}\left(\sum_{i=1}^d \|\mathbf{g}_{1:T,i}\|_2\right)^{1/2}\right)^{-1},$$

then by (4.2), we have

$$\mathbb{E}\left[\|\nabla f(\mathbf{x}_{\text{out}})\|_2^2\right] = O\left(\frac{\left(\sum_{i=1}^d \|\mathbf{g}_{1:T,i}\|_2\right)^{1/2}}{T^{3/4}} + \frac{d}{T}\right). \tag{4.4}$$

Note that the convergence rate given by (4.4) is related to the sum of gradient norms $\sum_{i=1}^d \|\mathbf{g}_{1:T,i}\|_2$. As we mentioned in Remark 4.6, when the stochastic gradients $\mathbf{g}_{1:T,i}$, $i = 1, \dots, d$ are sparse, we have $\sum_{i=1}^d \|\mathbf{g}_{1:T,i}\|_2 \ll \sqrt{dT}$ (Duchi et al. (2011)). More specifically, suppose $\sum_{i=1}^d \|\mathbf{g}_{1:T,i}\|_2 = O(d^s\sqrt{T})$ for some $0 \leq s \leq 1/2$. We have $\mathbb{E}[\|\nabla f(\mathbf{x}_{\text{out}})\|_2^2] = O(d^{s/2}/T^{1/2} + d/T)$. When $s = 1/2$, we have $\mathbb{E}[\|\nabla f(\mathbf{x}_{\text{out}})\|_2^2] = O(d^{1/4}/\sqrt{T} + d/T)$, which matches the rate $O(1/\sqrt{T})$ achieved by nonconvex SGD (Ghadimi & Lan, 2016), considering the dependence of $T$. It is worth noting that nonconvex SGD achieves $O(\max_{\mathbf{x},\xi} \|\nabla f(\mathbf{x}, ; \xi)\|_2/\sqrt{T}) = O(\sqrt{d/T})$ convergence rate under Assumption 4.1, thus the convergence rate of our algorithm is strictly better than that of SGD by a factor of $d^{1/4}$ when $T$ is large.

**Remark 4.8.** If we set $\alpha = 1/\sqrt{T}$ which is not related to $\sum_{i=1}^{d} \|\mathbf{g}_{1:T,i}\|_2$, then (4.2) suggests that

$$\mathbb{E}\Big[\|\nabla f(\mathbf{x}_{\text{out}})\|_2^2\Big] = O\bigg(\frac{1}{\sqrt{T}} + \frac{d}{T} + \frac{1}{T}\sum_{i=1}^{d}\|\mathbf{g}_{1:T,i}\|_2\bigg).$$

When $\sum_{i=1}^{d}\|\mathbf{g}_{1:T,i}\|_2 \ll \sqrt{dT}$ (Duchi et al., 2011; Reddi et al., 2018), we further have

$$\mathbb{E}\Big[\|\nabla f(\mathbf{x}_{\text{out}})\|_2^2\Big] = O\bigg(\frac{1}{\sqrt{T}} + \frac{d}{T} + \sqrt{\frac{d}{T}}\bigg),$$

which matches the convergence result in nonconvex SGD (Ghadimi & Lan, 2016) considering the dependence of $T$.

## 5 EXPERIMENTS

In this section, we empirically evaluate our proposed algorithm for training various modern deep learning models and test them on several standard benchmarks [6]. We show that for nonconvex loss functions in deep learning, our proposed algorithm still enjoys a fast convergence rate, while its generalization performance is as good as SGD with momentum and much better than existing adaptive gradient methods such as Adam and Amsgrad. All experiments are conducted on Amazon AWS p3.8xlarge servers which come with Intel Xeon E5 CPU and 4 NVIDIA Tesla V100 GPUs (16G RAM per GPU). All experiments are implemented in Pytorch version 0.4.1 within Python 3.6.4.

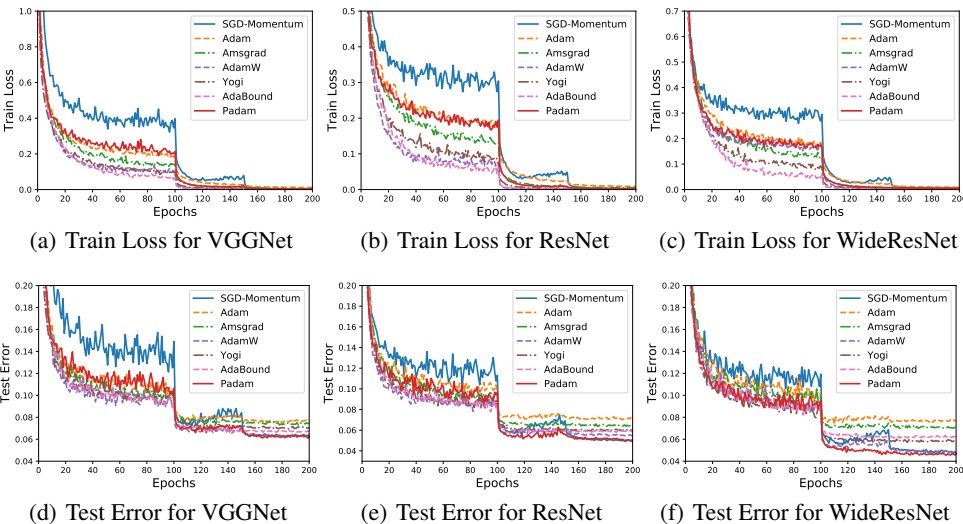

Figure 2: Train loss and test error (top-1) plots of three CNN architectures on CIFAR-10 dataset.

We compare Padam against several state-of-the-art algorithms, including: (1) SGD-momentum, (2) Adam (Kingma & Ba, 2015), (3) Amsgrad (Reddi et al., 2018), (4) AdamW (Loshchilov & Hutter, 2019) (5) Yogi (Zaheer et al., 2018) and (6) AdaBound (Luo et al., 2019) . We use several popular datasets for image classifications and language modeling: CIFAR-10 (Krizhevsky & Hinton, 2009), CIFAR-100 (Krizhevsky & Hinton, 2009), ImageNet dataset (ILSVRC2012) (Deng et al., 2009) and Penn Treebank dataset (Marcus et al., 1993). We adopt three popular CNN architectures to evaluate the performance of our proposed algorithms on image classification task: VGGNet-16 (Simonyan & Zisserman, 2014), Residual Neural Network (ResNet-18) (He et al., 2016), Wide Residual Network (WRN-16-4) (Zagoruyko & Komodakis, 2016). We test the language modeling task using 2-layer and 3-layer Long Short-Term Memory (LSTM) network (Hochreiter & Schmidhuber, 1997). We test CIFAR-10, CIFAR-100 and Penn Treebank tasks for 200 epochs. For all experiments, we use a fixed multi-stage learning rate decaying scheme: we decay the learning rate by 0.1 at the 100th and 150th epochs. We test ImageNet tasks for 100 epochs with similar multi-stage learning rate decaying

---

[6]The code is available in the anonymous link

scheme at the 30th, 60th and 80th epochs. We perform grid search on validation set to choose the best hyper-parameters for each algorithm. Details about the datasets, CNN architectures and all the specific model parameters used in the following experiments can be found in the supplementary materials.

## 5.1 EXPERIMENTAL RESULTS

We compare our proposed algorithm with other baselines on training the aforementioned three modern CNN architectures for image classification on the CIFAR-10, CIFAR-100 and also ImageNet datasets. Due to the paper length limit, we leave all our experimental results regarding CIFAR-100 dataset and all test result tables in the supplementary materials. Figure 2 plots the train loss and test error (top-1 error) against training epochs on the CIFAR-10 dataset. As we can see from the figure, at the early stage of the training process, (partially) adaptive gradient methods including Padam, make rapid progress lowing both the train loss and the test error, while SGD with momentum converges relatively slowly. After the first learning rate decaying at the 100-th epoch, different algorithms start to behave differently. SGD with momentum makes a huge

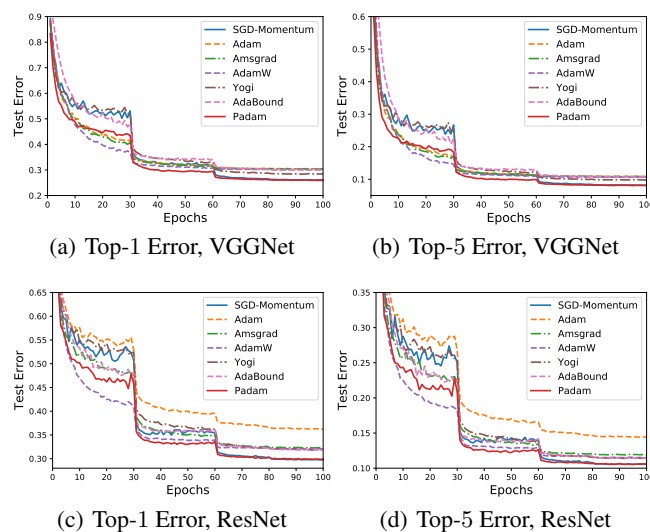

(a) Top-1 Error, VGGNet     (b) Top-5 Error, VGGNet

(c) Top-1 Error, ResNet     (d) Top-5 Error, ResNet

Figure 3: Top-1 and Top-5 test error for VGGNet and ResNet on ImageNet dataset.

drop while fully adaptive gradient methods (Adam and Amsgrad) start to generalize badly. Padam, on the other hand, maintains relatively good generalization performance and also holds the lead over other algorithms. After the second decaying at the 150-th epoch, Adam and Amsgrad basically lose all the power of traversing through the parameter space due to the "small learning dilemma", while the performance of SGD with momentum finally catches up with Padam. AdamW, Yogi and Ad-aBound indeed improve the performance compared with original Adam but the performance is still worse than Padam.

Overall we can see that Padam achieves the best of both worlds (i.e., Adam and SGD with momentum): it maintains faster convergence rate while also generalizing as well as SGD with momentum in the end. Figure 3 plots the Top-1 and Top-5 error against training epochs on the ImageNet dataset for both VGGNet and ResNet. We can see that on ImageNet, all methods behave similarly as in our CIFAR-10 experiments. Padam method again

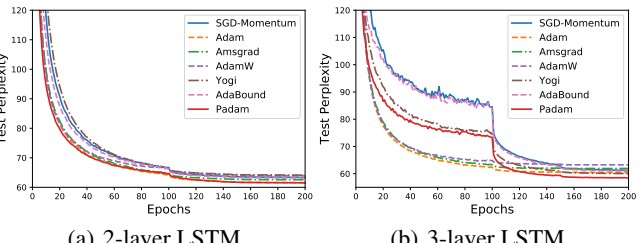

(a) 2-layer LSTM     (b) 3-layer LSTM

Figure 4: Test perplexity for 2-layer and 3-layer LSTM model on Penn Treebank dataset.

obtains the best from both worlds by achieving the fastest convergence while generalizing as well as SGD with momentum. Even though methods such as AdamW, Yogi and AdaBound have better performance than standard Adam, they still suffer from a big generalization gap on ImageNet dataset. Note that we did not conduct WideResNet experiment on Imagenet dataset due to GPU memory limits.

We also perform experiments on the language modeling tasks to test our proposed algorithm on Long Short-Term Memory (LSTM) network (Hochreiter & Schmidhuber, 1997), where adaptive gradient methods such as Adam are currently the mainstream optimizers for these tasks. Figure 4

plots the test perplexity against training epochs on the Penn Treebank dataset (Marcus et al., 1993) for both 2-layer LSTM and 3-layer LSTM models. We can observe that the differences on simpler 2-layer LSTM model is not very obvious but on more complicated 3-layer LSTM model, different algorithms have quite different optimizing behaviors. Even though Adam, Amsgrad and AdamW have faster convergence in the early stages, Padam achieves the best final test perplexity on this language modeling task for both of our experiments.

## 6 Conclusions and Future Work

In this paper, we proposed Padam, which unifies Adam/Amsgrad with SGD-momentum. With an appropriate choice of the partially adaptive parameter, we show that Padam can achieve the best from both worlds, i.e., maintaining fast convergence rate while closing the generalization gap. We also provide a theoretical analysis towards the convergence rate of Padam to a stationary point for stochastic nonconvex optimization.

While the empirical generalization performances achieved by Padam backup our hypothesis of "small learning rate dilemma", it is still unclear how learning rate decay affects the performance of adaptive gradient methods in theory. We leave it as an important future direction.

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

## A    PROOF OF THE MAIN THEORY

In this section, we provide a detailed version of proof of Theorem 4.3.

### A.1    PROOF OF THEOREM 4.3

Let $\mathbf{x}_0 = \mathbf{x}_1$ and

$$\mathbf{z}_t = \mathbf{x}_t + \frac{\beta_1}{1-\beta_1}(\mathbf{x}_t - \mathbf{x}_{t-1}) = \frac{1}{1-\beta_1}\mathbf{x}_t - \frac{\beta_1}{1-\beta_1}\mathbf{x}_{t-1}, \tag{A.1}$$

we have the following lemmas:

**Lemma A.1.** Let $\mathbf{z}_t$ be defined in (A.1). For $t \geq 2$, we have

$$\mathbf{z}_{t+1} - \mathbf{z}_t = \frac{\beta_1}{1-\beta_1}\Big[\mathbf{I} - \big(\alpha_t\widehat{\mathbf{V}}_t^{-p}\big)\big(\alpha_{t-1}\widehat{\mathbf{V}}_{t-1}^{-p}\big)^{-1}\Big](\mathbf{x}_{t-1} - \mathbf{x}_t) - \alpha_t\widehat{\mathbf{V}}_t^{-p}\mathbf{g}_t. \tag{A.2}$$

and

$$\mathbf{z}_{t+1} - \mathbf{z}_t = \frac{\beta_1}{1-\beta_1}\big(\alpha_{t-1}\widehat{\mathbf{V}}_{t-1}^{-p} - \alpha_t\widehat{\mathbf{V}}_t^{-p}\big)\mathbf{m}_{t-1} - \alpha_t\widehat{\mathbf{V}}_t^{-p}\mathbf{g}_t. \tag{A.3}$$

For $t = 1$, we have

$$\mathbf{z}_2 - \mathbf{z}_1 = -\alpha_1\widehat{\mathbf{V}}_1^{-p}\mathbf{g}_1. \tag{A.4}$$

**Lemma A.2.** Let $\mathbf{z}_t$ be defined in (A.1). For $t \geq 2$, we have

$$\|\mathbf{z}_{t+1} - \mathbf{z}_t\|_2 \leq \big\|\alpha\widehat{\mathbf{V}}_t^{-p}\mathbf{g}_t\big\|_2 + \frac{\beta_1}{1-\beta_1}\|\mathbf{x}_{t-1} - \mathbf{x}_t\|_2.$$

**Lemma A.3.** Let $\mathbf{z}_t$ be defined in (A.1). For $t \geq 2$, we have

$$\|\nabla f(\mathbf{z}_t) - \nabla f(\mathbf{x}_t)\|_2 \leq L\Big(\frac{\beta_1}{1-\beta_1}\Big) \cdot \|\mathbf{x}_t - \mathbf{x}_{t-1}\|_2.$$

**Lemma A.4.** Let $\widehat{\mathbf{v}}_t$ and $\mathbf{m}_t$ be as defined in Algorithm 1. Then under Assumption 4.1, we have $\|\nabla f(\mathbf{x})\|_\infty \leq G_\infty$, $\|\widehat{\mathbf{v}}_t\|_\infty \leq G_\infty^2$ and $\|\mathbf{m}_t\|_\infty \leq G_\infty$.

**Lemma A.5.** Suppose that $f$ has $G_\infty$-bounded stochastic gradient. Let $\beta_1, \beta_2$ be the weight parameters, $\alpha_t$, $t = 1, \ldots, T$ be the step sizes in Algorithm 1 and $q \in [\max\{4p - 1, 0\}, 1]$. We denote $\gamma = \beta_1/\beta_2^{2p}$. Suppose that $\alpha_t = \alpha$ and $\gamma \leq 1$, then under Assumption 4.1, we have the following two results:

$$\sum_{t=1}^T \alpha_t^2 \mathbb{E}\Big[\big\|\widehat{\mathbf{V}}_t^{-p}\mathbf{m}_t\big\|_2^2\Big] \leq \frac{T^{(1+q)/2}d^q\alpha^2(1-\beta_1)G_\infty^{(1+q-4p)}}{(1-\beta_2)^{2p}(1-\gamma)}\mathbb{E}\Big(\sum_{i=1}^d \|\mathbf{g}_{1:T,i}\|_2\Big)^{1-q},$$

and

$$\sum_{t=1}^T \alpha_t^2 \mathbb{E}\Big[\big\|\widehat{\mathbf{V}}_t^{-p}\mathbf{g}_t\big\|_2^2\Big] \leq \frac{T^{(1+q)/2}d^q\alpha^2 G_\infty^{(1+q-4p)}}{(1-\beta_2)^{2p}}\mathbb{E}\Big(\sum_{i=1}^d \|\mathbf{g}_{1:T,i}\|_2\Big)^{1-q}.$$

Now we are ready to prove Theorem 4.3.

*Proof of Theorem 4.3.* Since $f$ is $L$-smooth, we have:

$$f(\mathbf{z}_{t+1}) \leq f(\mathbf{z}_t) + \nabla f(\mathbf{z}_t)^\top(\mathbf{z}_{t+1} - \mathbf{z}_t) + \frac{L}{2}\|\mathbf{z}_{t+1} - \mathbf{z}_t\|_2^2$$

$$= f(\mathbf{z}_t) + \underbrace{\nabla f(\mathbf{x}_t)^\top(\mathbf{z}_{t+1} - \mathbf{z}_t)}_{I_1} + \underbrace{(\nabla f(\mathbf{z}_t) - \nabla f(\mathbf{x}_t))^\top(\mathbf{z}_{t+1} - \mathbf{z}_t)}_{I_2} + \underbrace{\frac{L}{2}\|\mathbf{z}_{t+1} - \mathbf{z}_t\|_2^2}_{I_3}$$

$$\tag{A.5}$$

In the following, we bound $I_1$, $I_2$ and $I_3$ separately.

**Bounding term $I_1$:** When $t = 1$, we have

$$\nabla f(\mathbf{x}_1)^\top (\mathbf{z}_2 - \mathbf{z}_1) = -\nabla f(\mathbf{x}_1)^\top \alpha_1 \widehat{\mathbf{V}}_t^{-p} \mathbf{g}_1. \tag{A.6}$$

For $t \geq 2$, we have

$$
\begin{aligned}
&\nabla f(\mathbf{x}_t)^\top (\mathbf{z}_{t+1} - \mathbf{z}_t) \\
&= \nabla f(\mathbf{x}_t)^\top \left[ \frac{\beta_1}{1 - \beta_1} \big( \alpha_{t-1} \widehat{\mathbf{V}}_{t-1}^{-p} - \alpha_t \widehat{\mathbf{V}}_t^{-p} \big) \mathbf{m}_{t-1} - \alpha_t \widehat{\mathbf{V}}_t^{-p} \mathbf{g}_t \right] \\
&= \frac{\beta_1}{1 - \beta_1} \nabla f(\mathbf{x}_t)^\top \big( \alpha_{t-1} \widehat{\mathbf{V}}_{t-1}^{-p} - \alpha_t \widehat{\mathbf{V}}_t^{-p} \big) \mathbf{m}_{t-1} - \nabla f(\mathbf{x}_t)^\top \alpha_t \widehat{\mathbf{V}}_t^{-p} \mathbf{g}_t, \tag{A.7}
\end{aligned}
$$

where the first equality holds due to (A.3) in Lemma A.1. For $\nabla f(\mathbf{x}_t)^\top (\alpha_{t-1} \widehat{\mathbf{V}}_{t-1}^{-p} - \alpha_t \widehat{\mathbf{V}}_t^{-p}) \mathbf{m}_{t-1}$ in (A.7), we have

$$
\begin{aligned}
\nabla f(\mathbf{x}_t)^\top (\alpha_{t-1} \widehat{\mathbf{V}}_{t-1}^{-p} - \alpha_t \widehat{\mathbf{V}}_t^{-p}) \mathbf{m}_{t-1} &\leq \|\nabla f(\mathbf{x}_t)\|_\infty \cdot \big\| \alpha_{t-1} \widehat{\mathbf{V}}_{t-1}^{-p} - \alpha_t \widehat{\mathbf{V}}_t^{-p} \big\|_{1,1} \cdot \|\mathbf{m}_{t-1}\|_\infty \\
&\leq G_\infty^2 \Big[ \big\| \alpha_{t-1} \widehat{\mathbf{V}}_{t-1}^{-p} \big\|_{1,1} - \big\| \alpha_t \widehat{\mathbf{V}}_t^{-p} \big\|_{1,1} \Big] \\
&= G_\infty^2 \Big[ \big\| \alpha_{t-1} \widehat{\mathbf{v}}_{t-1}^{-p} \big\|_1 - \big\| \alpha_t \widehat{\mathbf{v}}_t^{-p} \big\|_1 \Big]. \tag{A.8}
\end{aligned}
$$

The first inequality holds because for a positive diagonal matrix $\mathbf{A}$, we have $\mathbf{x}^\top \mathbf{A} \mathbf{y} \leq \|\mathbf{x}\|_\infty \cdot \|\mathbf{A}\|_{1,1} \cdot \|\mathbf{y}\|_\infty$. The second inequality holds due to $\alpha_{t-1} \widehat{\mathbf{V}}_{t-1}^{-p} \succeq \alpha_t \widehat{\mathbf{V}}_t^{-p} \succeq 0$. Next we bound $-\nabla f(\mathbf{x}_t)^\top \alpha_t \widehat{\mathbf{V}}_t^{-p} \mathbf{g}_t$. We have

$$
\begin{aligned}
&- \nabla f(\mathbf{x}_t)^\top \alpha_t \widehat{\mathbf{V}}_t^{-p} \mathbf{g}_t \\
&= -\nabla f(\mathbf{x}_t)^\top \alpha_{t-1} \widehat{\mathbf{V}}_{t-1}^{-p} \mathbf{g}_t - \nabla f(\mathbf{x}_t)^\top \big( \alpha_t \widehat{\mathbf{V}}_t^{-p} - \alpha_{t-1} \widehat{\mathbf{V}}_{t-1}^{-p} \big) \mathbf{g}_t \\
&\leq -\nabla f(\mathbf{x}_t)^\top \alpha_{t-1} \widehat{\mathbf{V}}_{t-1}^{-p} \mathbf{g}_t + \|\nabla f(\mathbf{x}_t)\|_\infty \cdot \big\| \alpha_t \widehat{\mathbf{V}}_t^{-p} - \alpha_{t-1} \widehat{\mathbf{V}}_{t-1}^{-p} \big\|_{1,1} \cdot \|\mathbf{g}_t\|_\infty \\
&\leq -\nabla f(\mathbf{x}_t)^\top \alpha_{t-1} \widehat{\mathbf{V}}_{t-1}^{-p} \mathbf{g}_t + G_\infty^2 \Big( \big\| \alpha_{t-1} \widehat{\mathbf{V}}_{t-1}^{-p} \big\|_{1,1} - \big\| \alpha_t \widehat{\mathbf{V}}_t^{-p} \big\|_{1,1} \Big) \\
&= -\nabla f(\mathbf{x}_t)^\top \alpha_{t-1} \widehat{\mathbf{V}}_{t-1}^{-p} \mathbf{g}_t + G_\infty^2 \Big( \big\| \alpha_{t-1} \widehat{\mathbf{v}}_{t-1}^{-p} \big\|_1 - \big\| \alpha_t \widehat{\mathbf{v}}_t^{-p} \big\|_1 \Big). \tag{A.9}
\end{aligned}
$$

The first inequality holds because for a positive diagonal matrix $\mathbf{A}$, we have $\mathbf{x}^\top \mathbf{A} \mathbf{y} \leq \|\mathbf{x}\|_\infty \cdot \|\mathbf{A}\|_{1,1} \cdot \|\mathbf{y}\|_\infty$. The second inequality holds due to $\alpha_{t-1} \widehat{\mathbf{V}}_{t-1}^{-p} \succeq \alpha_t \widehat{\mathbf{V}}_t^{-p} \succeq 0$. Substituting (A.8) and (A.9) into (A.7), we have

$$\nabla f(\mathbf{x}_t)^\top (\mathbf{z}_{t+1} - \mathbf{z}_t) \leq -\nabla f(\mathbf{x}_t)^\top \alpha_{t-1} \widehat{\mathbf{V}}_{t-1}^{-p} \mathbf{g}_t + \frac{1}{1 - \beta_1} G_\infty^2 \Big( \big\| \alpha_{t-1} \widehat{\mathbf{v}}_{t-1}^{-p} \big\|_1 - \big\| \alpha_t \widehat{\mathbf{v}}_t^{-p} \big\|_1 \Big). \tag{A.10}$$

**Bounding term $I_2$:** For $t \geq 1$, we have

$$
\begin{aligned}
&\big( \nabla f(\mathbf{z}_t) - \nabla f(\mathbf{x}_t) \big)^\top (\mathbf{z}_{t+1} - \mathbf{z}_t) \\
&\leq \big\| \nabla f(\mathbf{z}_t) - \nabla f(\mathbf{x}_t) \big\|_2 \cdot \|\mathbf{z}_{t+1} - \mathbf{z}_t\|_2 \\
&\leq \left( \big\| \alpha_t \widehat{\mathbf{V}}_t^{-p} \mathbf{g}_t \big\|_2 + \frac{\beta_1}{1 - \beta_1} \|\mathbf{x}_{t-1} - \mathbf{x}_t\|_2 \right) \cdot \frac{\beta_1}{1 - \beta_1} \cdot L \|\mathbf{x}_t - \mathbf{x}_{t-1}\|_2 \\
&= L \frac{\beta_1}{1 - \beta_1} \big\| \alpha_t \widehat{\mathbf{V}}_t^{-p} \mathbf{g}_t \big\|_2 \cdot \|\mathbf{x}_t - \mathbf{x}_{t-1}\|_2 + L \left( \frac{\beta_1}{1 - \beta_1} \right)^2 \|\mathbf{x}_t - \mathbf{x}_{t-1}\|_2^2 \\
&\leq L \big\| \alpha_t \widehat{\mathbf{V}}_t^{-p} \mathbf{g}_t \big\|_2^2 + 2L \left( \frac{\beta_1}{1 - \beta_1} \right)^2 \|\mathbf{x}_t - \mathbf{x}_{t-1}\|_2^2, \tag{A.11}
\end{aligned}
$$

where the second inequality holds because of Lemma A.1 and Lemma A.2, the last inequality holds due to Young's inequality.

**Bounding term $I_3$:** For $t \geq 1$, we have

$$
\begin{aligned}
\frac{L}{2} \|\mathbf{z}_{t+1} - \mathbf{z}_t\|_2^2 &\leq \frac{L}{2} \left[ \big\| \alpha_t \widehat{\mathbf{V}}_t^{-p} \mathbf{g}_t \big\|_2 + \frac{\beta_1}{1 - \beta_1} \|\mathbf{x}_{t-1} - \mathbf{x}_t\|_2 \right]^2 \\
&\leq L \big\| \alpha_t \widehat{\mathbf{V}}_t^{-p} \mathbf{g}_t \big\|_2^2 + 2L \left( \frac{\beta_1}{1 - \beta_1} \right)^2 \|\mathbf{x}_{t-1} - \mathbf{x}_t\|_2^2. \tag{A.12}
\end{aligned}
$$

The first inequality is obtained by introducing Lemma A.1.

For $t = 1$, substituting (A.6), (A.11) and (A.12) into (A.5), taking expectation and rearranging terms, we have

$$
\begin{aligned}
&\mathbb{E}[f(\mathbf{z}_2) - f(\mathbf{z}_1)] \\
&\leq \mathbb{E}\left[ -\nabla f(\mathbf{x}_1)^\top \alpha_1 \widehat{\mathbf{V}}_1^{-p} \mathbf{g}_1 + 2L \big\| \alpha_1 \widehat{\mathbf{V}}_1^{-p} \mathbf{g}_1 \big\|_2^2 + 4L\left(\frac{\beta_1}{1-\beta_1}\right)^2 \|\mathbf{x}_1 - \mathbf{x}_0\|_2^2 \right] \\
&= \mathbb{E}[-\nabla f(\mathbf{x}_1)^\top \alpha_1 \widehat{\mathbf{V}}_1^{-p} \mathbf{g}_1 + 2L \big\| \alpha_1 \widehat{\mathbf{V}}_1^{-p} \mathbf{g}_1 \big\|_2^2] \\
&\leq \mathbb{E}[d\alpha_1 G_\infty^{2-2p} + 2L \big\| \alpha_1 \widehat{\mathbf{V}}_1^{-p} \mathbf{g}_1 \big\|_2^2],
\end{aligned} \tag{A.13}
$$

where the last inequality holds because

$$
-\nabla f(\mathbf{x}_1)^\top \widehat{\mathbf{V}}_1^{-p} \mathbf{g}_1 \leq d \cdot \|\nabla f(\mathbf{x}_1)\|_\infty \cdot \|\widehat{\mathbf{V}}_1^{-p} \mathbf{g}_1\|_\infty \leq d \cdot G_\infty \cdot G_\infty^{1-2p} = dG_\infty^{2-2p}.
$$

For $t \geq 2$, substituting (A.10), (A.11) and (A.12) into (A.5), taking expectation and rearranging terms, we have

$$
\begin{aligned}
&\mathbb{E}\left[ f(\mathbf{z}_{t+1}) + \frac{G_\infty^2 \big\| \alpha_t \widehat{\mathbf{v}}_t^{-p} \big\|_1}{1-\beta_1} - \left( f(\mathbf{z}_t) + \frac{G_\infty^2 \big\| \alpha_{t-1} \widehat{\mathbf{v}}_{t-1}^{-p} \big\|_1}{1-\beta_1} \right) \right] \\
&\leq \mathbb{E}\left[ -\nabla f(\mathbf{x}_t)^\top \alpha_{t-1} \widehat{\mathbf{V}}_{t-1}^{-p} \mathbf{g}_t + 2L \big\| \alpha_t \widehat{\mathbf{V}}_t^{-p} \mathbf{g}_t \big\|_2^2 + 4L\left(\frac{\beta_1}{1-\beta_1}\right)^2 \|\mathbf{x}_t - \mathbf{x}_{t-1}\|_2^2 \right] \\
&= \mathbb{E}\left[ -\nabla f(\mathbf{x}_t)^\top \alpha_{t-1} \widehat{\mathbf{V}}_{t-1}^{-p} \nabla f(\mathbf{x}_t) + 2L \big\| \alpha_t \widehat{\mathbf{V}}_t^{-p} \mathbf{g}_t \big\|_2^2 + 4L\left(\frac{\beta_1}{1-\beta_1}\right)^2 \big\| \alpha_{t-1} \widehat{\mathbf{V}}_{t-1}^{-p} \mathbf{m}_{t-1} \big\|_2^2 \right] \\
&\leq \mathbb{E}\left[ -\alpha_{t-1} \big\| \nabla f(\mathbf{x}_t) \big\|_2^2 (G_\infty^{2p})^{-1} + 2L \big\| \alpha_t \widehat{\mathbf{V}}_t^{-p} \mathbf{g}_t \big\|_2^2 + 4L\left(\frac{\beta_1}{1-\beta_1}\right)^2 \big\| \alpha_{t-1} \widehat{\mathbf{V}}_{t-1}^{-p} \mathbf{m}_{t-1} \big\|_2^2 \right],
\end{aligned} \tag{A.14}
$$

where the equality holds because $\mathbb{E}[\mathbf{g}_t] = \nabla f(\mathbf{x}_t)$ conditioned on $\nabla f(\mathbf{x}_t)$ and $\widehat{\mathbf{V}}_{t-1}^{-p}$, the second inequality holds because of Lemma A.4. Telescoping (A.14) for $t = 2$ to $T$ and adding with (A.13), we have

$$
\begin{aligned}
&(G_\infty^{2p})^{-1} \sum_{t=2}^{T} \alpha_{t-1} \mathbb{E} \big\| \nabla f(\mathbf{x}_t) \big\|_2^2 \\
&\leq \mathbb{E}\left[ f(\mathbf{z}_1) + \frac{G_\infty^2 \big\| \alpha_1 \widehat{\mathbf{v}}_1^{-p} \big\|_1}{1-\beta_1} + d\alpha_1 G_\infty^{2-2p} - \left( f(\mathbf{z}_{T+1}) + \frac{G_\infty^2 \big\| \alpha_T \widehat{\mathbf{v}}_T^{-p} \big\|_1}{1-\beta_1} \right) \right] \\
&\quad + 2L \sum_{t=1}^{T} \mathbb{E} \big\| \alpha_t \widehat{\mathbf{V}}_t^{-p} \mathbf{g}_t \big\|_2^2 + 4L\left(\frac{\beta_1}{1-\beta_1}\right)^2 \sum_{t=2}^{T} \mathbb{E}\left[ \big\| \alpha_{t-1} \widehat{\mathbf{V}}_{t-1}^{-p} \mathbf{m}_{t-1} \big\|_2^2 \right] \\
&\leq \mathbb{E}\left[ \Delta f + \frac{G_\infty^2 \big\| \alpha_1 \widehat{\mathbf{v}}_1^{-p} \big\|_1}{1-\beta_1} + d\alpha_1 G_\infty^{2-2p} \right] + 2L \sum_{t=1}^{T} \mathbb{E} \big\| \alpha_t \widehat{\mathbf{V}}_t^{-p} \mathbf{g}_t \big\|_2^2 \\
&\quad + 4L\left(\frac{\beta_1}{1-\beta_1}\right)^2 \sum_{t=1}^{T} \mathbb{E}\left[ \big\| \alpha_t \widehat{\mathbf{V}}_t^{-p} \mathbf{m}_t \big\|_2^2 \right].
\end{aligned} \tag{A.15}
$$

By Lemma A.5, we have

$$
\sum_{t=1}^{T} \alpha_t^2 \mathbb{E}\left[ \big\| \widehat{\mathbf{V}}_t^{-p} \mathbf{m}_t \big\|_2^2 \right] \leq \frac{T^{(1+q)/2} d^q \alpha^2 (1-\beta_1) G_\infty^{(1+q-4p)}}{(1-\beta_2)^{2p}(1-\gamma)} \mathbb{E}\left( \sum_{i=1}^{d} \|\mathbf{g}_{1:T,i}\|_2 \right)^{1-q}, \tag{A.16}
$$

where $\gamma = \beta_1/\beta_2^{2p}$. We also have

$$
\sum_{t=1}^{T} \alpha_t^2 \mathbb{E}\left[ \big\| \widehat{\mathbf{V}}_t^{-p} \mathbf{g}_t \big\|_2^2 \right] \leq \frac{T^{(1+q)/2} d^q \alpha^2 G_\infty^{(1+q-4p)}}{(1-\beta_2)^{2p}} \mathbb{E}\left( \sum_{i=1}^{d} \|\mathbf{g}_{1:T,i}\|_2 \right)^{1-q}. \tag{A.17}
$$

Substituting (A.16) and (A.17) into (A.15), and rearranging (A.15), we have

$$
\mathbb{E}\|\nabla f(\mathbf{x}_{\text{out}})\|_2^2
$$

$$
= \frac{1}{\sum_{t=2}^{T} \alpha_{t-1}} \sum_{t=2}^{T} \alpha_{t-1}\mathbb{E}\big\|\nabla f(\mathbf{x}_t)\big\|_2^2
$$

$$
\leq \frac{G_\infty^{2p}}{\sum_{t=2}^{T} \alpha_{t-1}}\mathbb{E}\bigg[\Delta f + \frac{G_\infty^2\big\|\alpha_1\widehat{\mathbf{v}}_1^{-p}\big\|_1}{1 - \beta_1} + d\alpha_1 G_\infty^{2-2p}\bigg]
$$

$$
+ \frac{2LG_\infty^{2p}}{\sum_{t=2}^{T} \alpha_{t-1}}\frac{T^{(1+q)/2}d^q\alpha^2 G_\infty^{(1+q-4p)}}{(1 - \beta_2)^{2p}}\mathbb{E}\bigg(\sum_{i=1}^{d}\|\mathbf{g}_{1:T,i}\|_2\bigg)^{1-q}
$$

$$
+ \frac{4LG_\infty^{2p}}{\sum_{t=2}^{T} \alpha_{t-1}}\bigg(\frac{\beta_1}{1 - \beta_1}\bigg)^2\frac{T^{(1+q)/2}d^q\alpha^2(1 - \beta_1)G_\infty^{(1+q-4p)}}{(1 - \beta_2)^{2p}(1 - \gamma)}\mathbb{E}\bigg(\sum_{i=1}^{d}\|\mathbf{g}_{1:T,i}\|_2\bigg)^{1-q}
$$

$$
\leq \frac{1}{T\alpha}2G_\infty^{2p}\Delta f + \frac{4}{T}\bigg(\frac{G_\infty^{2+2p}\mathbb{E}\big\|\widehat{\mathbf{v}}_1^{-p}\big\|_1}{1 - \beta_1} + dG_\infty^2\bigg)
$$

$$
+ \frac{d^q\alpha}{T^{(1-q)/2}}\mathbb{E}\bigg(\sum_{i=1}^{d}\|\mathbf{g}_{1:T,i}\|_2\bigg)^{1-q}\bigg(\frac{4LG_\infty^{1+q-2p}}{(1 - \beta_2)^{2p}} + \frac{8LG_\infty^{1+q-2p}(1 - \beta_1)}{(1 - \beta_2)^{2p}(1 - \gamma)}\bigg(\frac{\beta_1}{1 - \beta_1}\bigg)^2\bigg),
$$

$$
\text{(A.18)}
$$

where the second inequality holds because $\alpha_t = \alpha$. Rearranging (A.18), we obtain

$$
\mathbb{E}\|\nabla f(\mathbf{x}_{\text{out}})\|_2^2 \leq \frac{M_1}{T\alpha} + \frac{M_2 d}{T} + \frac{\alpha d^q M_3}{T^{(1-q)/2}}\mathbb{E}\bigg(\sum_{i=1}^{d}\|\mathbf{g}_{1:T,i}\|_2\bigg)^{1-q},
$$

where $\{M_i\}_{i=1}^3$ are defined in Theorem 4.3. This completes the proof. $\qquad\square$

## A.2 Proof of Corollary 4.5

*Proof of Corollary 4.5.* From Theorem 4.3, let $p \in [0, 1/4]$, we have $q \in [0, 1]$. Setting $q = 0$, we have

$$
\mathbb{E}\Big[\big\|\nabla f(\mathbf{x}_{\text{out}})\big\|_2^2\Big] \leq \frac{M_1}{T\alpha} + \frac{M_2 \cdot d}{T} + \frac{M_3'\alpha}{\sqrt{T}}\mathbb{E}\bigg(\sum_{i=1}^{d}\|\mathbf{g}_{1:T,i}\|_2\bigg),
$$

where $M_1$ and $M_2$ are defined in Theorem 4.3 and $M_3'$ is defined in Corollary 4.5. This completes the proof. $\qquad\square$

## B Proof of Technical Lemmas

### B.1 Proof of Lemma A.1

*Proof.* By definition, we have

$$
\mathbf{z}_{t+1} = \mathbf{x}_{t+1} + \frac{\beta_1}{1 - \beta_1}(\mathbf{x}_{t+1} - \mathbf{x}_t) = \frac{1}{1 - \beta_1}\mathbf{x}_{t+1} - \frac{\beta_1}{1 - \beta_1}\mathbf{x}_t.
$$

Then we have

$$
\mathbf{z}_{t+1} - \mathbf{z}_t = \frac{1}{1 - \beta_1}(\mathbf{x}_{t+1} - \mathbf{x}_t) - \frac{\beta_1}{1 - \beta_1}(\mathbf{x}_t - \mathbf{x}_{t-1})
$$

$$
= \frac{1}{1 - \beta_1}\big(-\alpha_t\widehat{\mathbf{V}}_t^{-p}\mathbf{m}_t\big) + \frac{\beta_1}{1 - \beta_1}\alpha_{t-1}\widehat{\mathbf{V}}_{t-1}^{-p}\mathbf{m}_{t-1}.
$$

The equities above are based on definition. Then we have

$$
\begin{aligned}
\mathbf{z}_{t+1} - \mathbf{z}_t &= \frac{-\alpha_t \widehat{\mathbf{V}}_t^{-p}}{1 - \beta_1} \Big[ \beta_1 \mathbf{m}_{t-1} + (1 - \beta_1)\mathbf{g}_t \Big] + \frac{\beta_1}{1 - \beta_1} \alpha_{t-1} \widehat{\mathbf{V}}_{t-1}^{-p} \mathbf{m}_{t-1} \\
&= \frac{\beta_1}{1 - \beta_1} \mathbf{m}_{t-1} \big( \alpha_{t-1} \widehat{\mathbf{V}}_{t-1}^{-p} - \alpha_t \widehat{\mathbf{V}}_t^{-p} \big) - \alpha_t \widehat{\mathbf{V}}_t^{-p} \mathbf{g}_t \\
&= \frac{\beta_1}{1 - \beta_1} \alpha_{t-1} \widehat{\mathbf{V}}_{t-1}^{-p} \mathbf{m}_{t-1} \Big[ \mathbf{I} - \big( \alpha_t \widehat{\mathbf{V}}_t^{-p} \big) \big( \alpha_{t-1} \widehat{\mathbf{V}}_{t-1}^{-p} \big)^{-1} \Big] - \alpha_t \widehat{\mathbf{V}}_t^{-p} \mathbf{g}_t \\
&= \frac{\beta_1}{1 - \beta_1} \Big[ \mathbf{I} - \big( \alpha_t \widehat{\mathbf{V}}_t^{-p} \big) \big( \alpha_{t-1} \widehat{\mathbf{V}}_{t-1}^{-p} \big)^{-1} \Big] (\mathbf{x}_{t-1} - \mathbf{x}_t) - \alpha_t \widehat{\mathbf{V}}_t^{-p} \mathbf{g}_t .
\end{aligned}
$$

The equalities above follow by combining the like terms. $\qquad \square$

## B.2 PROOF OF LEMMA A.2

*Proof.* By Lemma A.1, we have

$$
\begin{aligned}
\| \mathbf{z}_{t+1} - \mathbf{z}_t \|_2 &= \left\| \frac{\beta_1}{1 - \beta_1} \Big[ \mathbf{I} - (\alpha_t \widehat{\mathbf{V}}_t^{-p})(\alpha_{t-1} \widehat{\mathbf{V}}_{t-1}^{-p})^{-1} \Big] (\mathbf{x}_{t-1} - \mathbf{x}_t) - \alpha_t \widehat{\mathbf{V}}_t^{-p} \mathbf{g}_t \right\|_2 \\
&\leq \frac{\beta_1}{1 - \beta_1} \left\| \mathbf{I} - (\alpha_t \widehat{\mathbf{V}}_t^{-p})(\alpha_{t-1} \widehat{\mathbf{V}}_{t-1}^{-p})^{-1} \right\|_{\infty,\infty} \cdot \| \mathbf{x}_{t-1} - \mathbf{x}_t \|_2 + \left\| \alpha \widehat{\mathbf{V}}_t^{-p} \mathbf{g}_t \right\|_2 ,
\end{aligned}
$$

where the inequality holds because the term $\beta_1/(1 - \beta_1)$ is positive, and triangle inequality. Considering that $\alpha_t \widehat{\mathbf{v}}_{t,j}^{-p} \leq \alpha_{t-1} \widehat{\mathbf{v}}_{t-1,j}^{-p}$, when $p > 0$, we have $\left\| \mathbf{I} - (\alpha_t \widehat{\mathbf{V}}_t^{-p})(\alpha_{t-1} \widehat{\mathbf{V}}_{t-1}^{-p})^{-1} \right\|_{\infty,\infty} \leq 1$. With that fact, the term above can be bound as:

$$
\| \mathbf{z}_{t+1} - \mathbf{z}_t \|_2 \leq \left\| \alpha \widehat{\mathbf{V}}_t^{-p} \mathbf{g}_t \right\|_2 + \frac{\beta_1}{1 - \beta_1} \| \mathbf{x}_{t-1} - \mathbf{x}_t \|_2 .
$$

This completes the proof. $\qquad \square$

## B.3 PROOF OF LEMMA A.3

*Proof.* For term $\| \nabla f(\mathbf{z}_t) - \nabla f(\mathbf{x}_t) \|_2$, we have:

$$
\begin{aligned}
\| \nabla f(\mathbf{z}_t) - \nabla f(\mathbf{x}_t) \|_2 &\leq L \| \mathbf{z}_t - \mathbf{x}_t \|_2 \\
&\leq L \left\| \frac{\beta_1}{1 - \beta_1} (\mathbf{x}_t - \mathbf{x}_{t-1}) \right\|_2 \\
&\leq L \Big( \frac{\beta_1}{1 - \beta_1} \Big) \cdot \| \mathbf{x}_t - \mathbf{x}_{t-1} \|_2 ,
\end{aligned}
$$

where the last inequality holds because the term $\beta_1/(1 - \beta_1)$ is positive. $\qquad \square$

## B.4 PROOF OF LEMMA A.4

*Proof of Lemma A.4.* Since $f$ has $G_\infty$-bounded stochastic gradient, for any $\mathbf{x}$ and $\xi$, $\| \nabla f(\mathbf{x}; \xi) \|_\infty \leq G_\infty$. Thus, we have

$$
\| \nabla f(\mathbf{x}) \|_\infty = \| \mathbb{E}_\xi \nabla f(\mathbf{x}; \xi) \|_\infty \leq \mathbb{E}_\xi \| \nabla f(\mathbf{x}; \xi) \|_\infty \leq G_\infty .
$$

Next we bound $\| \mathbf{m}_t \|_\infty$. We have $\| \mathbf{m}_0 \| = 0 \leq G_\infty$. Suppose that $\| \mathbf{m}_t \|_\infty \leq G_\infty$, then for $\mathbf{m}_{t+1}$, we have

$$
\begin{aligned}
\| \mathbf{m}_{t+1} \|_\infty &= \| \beta_1 \mathbf{m}_t + (1 - \beta_1)\mathbf{g}_{t+1} \|_\infty \\
&\leq \beta_1 \| \mathbf{m}_t \|_\infty + (1 - \beta_1) \| \mathbf{g}_{t+1} \|_\infty \\
&\leq \beta_1 G_\infty + (1 - \beta_1) G_\infty \\
&= G_\infty .
\end{aligned}
$$

Thus, for any $t \geq 0$, we have $\|\mathbf{m}_t\|_\infty \leq G_\infty$. Finally we bound $\|\widehat{\mathbf{v}}_t\|_\infty$. First we have $\|\mathbf{v}_0\|_\infty = \|\widehat{\mathbf{v}}_0\|_\infty = 0 \leq G_\infty^2$. Suppose that $\|\widehat{\mathbf{v}}_t\|_\infty \leq G_\infty^2$ and $\|\mathbf{v}_t\|_\infty \leq G_\infty^2$. Note that we have

$$
\begin{aligned}
\|\mathbf{v}_{t+1}\|_\infty &= \|\beta_2 \mathbf{v}_t + (1-\beta_2)\mathbf{g}_{t+1}^2\|_\infty \\
&\leq \beta_2 \|\mathbf{v}_t\|_\infty + (1-\beta_2)\|\mathbf{g}_{t+1}^2\|_\infty \\
&\leq \beta_2 G_\infty^2 + (1-\beta_2)G_\infty^2 \\
&= G_\infty^2,
\end{aligned}
$$

and by definition, we have $\|\widehat{\mathbf{v}}_{t+1}\|_\infty = \max\{\|\widehat{\mathbf{v}}_t\|_\infty, \|\mathbf{v}_{t+1}\|_\infty\} \leq G_\infty^2$. Thus, for any $t \geq 0$, we have $\|\widehat{\mathbf{v}}_t\|_\infty \leq G_\infty^2$. $\qquad\square$

## B.5 PROOF OF LEMMA A.5

*Proof.* Recall that $\widehat{v}_{t,j}, m_{t,j}, g_{t,j}$ denote the $j$-th coordinate of $\widehat{\mathbf{v}}_t, \mathbf{m}_t$ and $\mathbf{g}_t$. We have

$$
\begin{aligned}
\alpha_t^2 \mathbb{E}\Big[\|\widehat{\mathbf{V}}_t^{-p}\mathbf{m}_t\|_2^2\Big] &= \alpha_t^2 \mathbb{E}\bigg[\sum_{i=1}^d \frac{m_{t,i}^2}{\widehat{v}_{t,i}^{2p}}\bigg] \\
&\leq \alpha_t^2 \mathbb{E}\bigg[\sum_{i=1}^d \frac{m_{t,i}^2}{v_{t,i}^{2p}}\bigg] \\
&= \alpha_t^2 \mathbb{E}\bigg[\sum_{i=1}^d \frac{(\sum_{j=1}^t (1-\beta_1)\beta_1^{t-j} g_{j,i})^2}{(\sum_{j=1}^t (1-\beta_2)\beta_2^{t-j} g_{j,i}^2)^{2p}}\bigg],
\end{aligned}
\tag{B.1}
$$

where the first inequality holds because $\widehat{v}_{t,i} \geq v_{t,i}$. Next we have

$$
\begin{aligned}
&\alpha_t^2 \mathbb{E}\bigg[\sum_{i=1}^d \frac{(\sum_{j=1}^t (1-\beta_1)\beta_1^{t-j} g_{j,i})^2}{(\sum_{j=1}^t (1-\beta_2)\beta_2^{t-j} g_{j,i}^2)^{2p}}\bigg] \\
&\leq \frac{\alpha_t^2(1-\beta_1)^2}{(1-\beta_2)^{2p}}\mathbb{E}\bigg[\sum_{i=1}^d \frac{(\sum_{j=1}^t \beta_1^{t-j}|g_{j,i}|^{(1+q-4p)})(\sum_{j=1}^t \beta_1^{t-j}|g_{j,i}|^{(1-q+4p)})}{(\sum_{j=1}^t \beta_2^{t-j} g_{j,i}^2)^{2p}}\bigg] \\
&\leq \frac{\alpha_t^2(1-\beta_1)^2}{(1-\beta_2)^{2p}}\mathbb{E}\bigg[\sum_{i=1}^d \frac{(\sum_{j=1}^t \beta_1^{t-j} G_\infty^{(1+q-4p)})(\sum_{j=1}^t \beta_1^{t-j}|g_{j,i}|^{(1-q+4p)})}{(\sum_{j=1}^t \beta_2^{t-j} g_{j,i}^2)^{2p}}\bigg] \\
&\leq \frac{\alpha_t^2(1-\beta_1) G_\infty^{(1+q-4p)}}{(1-\beta_2)^{2p}}\mathbb{E}\bigg[\sum_{i=1}^d \frac{\sum_{j=1}^t \beta_1^{t-j}|g_{j,i}|^{(1-q+4p)}}{(\sum_{j=1}^t \beta_2^{t-j} g_{j,i}^2)^{2p}}\bigg],
\end{aligned}
\tag{B.2}
$$

where the first inequality holds due to Cauchy inequality, the second inequality holds because $|g_{j,i}| \leq G_\infty$, the last inequality holds because $\sum_{j=1}^t \beta_1^{t-j} \leq (1-\beta_1)^{-1}$. Note that

$$
\sum_{i=1}^d \frac{\sum_{j=1}^t \beta_1^{t-j}|g_{j,i}|^{(1-q+4p)}}{(\sum_{j=1}^t \beta_2^{t-j} g_{j,i}^2)^{2p}} \leq \sum_{i=1}^d \sum_{j=1}^t \frac{\beta_1^{t-j}|g_{j,i}|^{(1-q+4p)}}{(\beta_2^{t-j} g_{j,i}^2)^{2p}} = \sum_{i=1}^d \sum_{j=1}^t \gamma^{t-j}|g_{j,i}|^{1-q},
\tag{B.3}
$$

where the equality holds due to the definition of $\gamma$. Substituting (B.2) and (B.3) into (B.1), we have

$$
\alpha_t^2 \mathbb{E}\Big[\|\widehat{\mathbf{V}}_t^{-p}\mathbf{m}_t\|_2^2\Big] \leq \frac{\alpha_t^2(1-\beta_1) G_\infty^{(1+q-4p)}}{(1-\beta_2)^{2p}}\mathbb{E}\bigg[\sum_{i=1}^d \sum_{j=1}^t \gamma^{t-j}|g_{j,i}|^{1-q}\bigg].
\tag{B.4}
$$

Telescoping (B.4) for $t=1$ to $T$, we have

$$
\begin{aligned}
\sum_{t=1}^T \alpha_t^2 \mathbb{E}\Big[\|\widehat{\mathbf{V}}_t^{-p}\mathbf{m}_t\|_2^2\Big] &\leq \frac{\alpha^2(1-\beta_1) G_\infty^{(1+q-4p)}}{(1-\beta_2)^{2p}}\mathbb{E}\bigg[\sum_{t=1}^T \sum_{i=1}^d \sum_{j=1}^t \gamma^{t-j}|g_{j,i}|^{1-q}\bigg] \\
&= \frac{\alpha^2(1-\beta_1) G_\infty^{(1+q-4p)}}{(1-\beta_2)^{2p}}\mathbb{E}\bigg[\sum_{i=1}^d \sum_{j=1}^T |g_{j,i}|^{1-q} \sum_{t=j}^T \gamma^{t-j}\bigg] \\
&\leq \frac{\alpha^2(1-\beta_1) G_\infty^{(1+q-4p)}}{(1-\beta_2)^{2p}(1-\gamma)}\mathbb{E}\bigg[\sum_{i=1}^d \sum_{j=1}^T |g_{j,i}|^{1-q}\bigg].
\end{aligned}
\tag{B.5}
$$

Finally, we have

$$
\sum_{i=1}^{d}\sum_{j=1}^{T}|g_{j,i}|^{1-q} \leq \sum_{i=1}^{d}\Big(\sum_{j=1}^{T}g_{j,i}^{2}\Big)^{(1-q)/2} \cdot T^{(1+q)/2}
$$
$$
= T^{(1+q)/2}\sum_{i=1}^{d}\|\mathbf{g}_{1:T,i}\|_{2}^{1-q}
$$
$$
\leq T^{(1+q)/2}d^{q}\bigg(\sum_{i=1}^{d}\|\mathbf{g}_{1:T,i}\|_{2}\bigg)^{1-q}, \tag{B.6}
$$

where the first and second inequalities hold due to Hölder's inequality. Substituting (B.6) into (B.5), we have

$$
\sum_{t=1}^{T}\alpha_{t}^{2}\mathbb{E}\Big[\|\widehat{\mathbf{V}}_{t}^{-p}\mathbf{m}_{t}\|_{2}^{2}\Big] \leq \frac{T^{(1+q)/2}d^{q}\alpha^{2}(1-\beta_{1})G_{\infty}^{(1+q-4p)}}{(1-\beta_{2})^{2p}(1-\gamma)}\mathbb{E}\bigg(\sum_{i=1}^{d}\|\mathbf{g}_{1:T,i}\|_{2}\bigg)^{1-q}.
$$

Specifically, taking $\beta_{1}=0$, we have $\mathbf{m}_{t}=\mathbf{g}_{t}$, then

$$
\sum_{t=1}^{T}\alpha_{t}^{2}\mathbb{E}\Big[\|\widehat{\mathbf{V}}_{t}^{-p}\mathbf{g}_{t}\|_{2}^{2}\Big] \leq \frac{T^{(1+q)/2}d^{q}\alpha^{2}G_{\infty}^{(1+q-4p)}}{(1-\beta_{2})^{2p}}\mathbb{E}\bigg(\sum_{i=1}^{d}\|\mathbf{g}_{1:T,i}\|_{2}\bigg)^{1-q}.
$$

$\square$

## C  EXPERIMENT DETAILS

### C.1  DATASETS

We use several popular datasets for image classifications.

- CIFAR-10 (Krizhevsky & Hinton, 2009): it consists of a training set of $50,000$ $32 \times 32$ color images from 10 classes, and also $10,000$ test images.
- CIFAR-100 (Krizhevsky & Hinton, 2009): it is similar to CIFAR-10 but contains 100 image classes with 600 images for each.
- ImageNet dataset (ILSVRC2012) (Deng et al., 2009): ILSVRC2012 contains 1.28 million training images, and $50k$ validation images over 1000 classes.

In addition, we adopt Penn Treebank (PTB) dataset (Marcus et al., 1993), which is widely used in Natural Language Processing (NLP) research. Note that word-level PTB dataset does not contain capital letters, numbers, and punctuation. Models are evaluated based on the perplexity metric (lower is better).

### C.2  ARCHITECTURES

**VGGNet** (Simonyan & Zisserman, 2014): We use a modified VGG-16 architecture for this experiment. The VGG-16 network uses only $3 \times 3$ convolutional layers stacked on top of each other for increasing depth and adopts max pooling to reduce volume size. Finally, two fully-connected layers [7] are followed by a softmax classifier.

**ResNet** (He et al., 2016): Residual Neural Network (ResNet) introduces a novel architecture with "skip connections" and features a heavy use of batch normalization. Such skip connections are also known as gated units or gated recurrent units and have a strong similarity to recent successful elements applied in RNNs. We use ResNet-18 for this experiment, which contains 2 blocks for each type of basic convolutional building blocks in He et al. (2016).

---

[7]For CIFAR experiments, we change the ending two fully-connected layers from 2048 nodes to 512 nodes. For ImageNet experiments, we use batch normalized version (vgg16_bn) provided in Pytorch official package

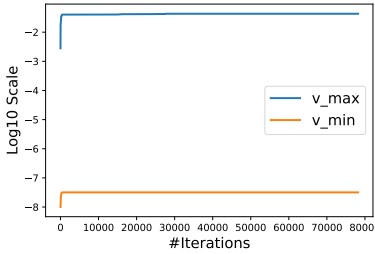

Figure 5: Plot of max and min values across all coordinates of $\widehat{\mathbf{v}}_t$ against the iteration number for ResNet model on CIFAR10 dataset.

**Wide ResNet** (Zagoruyko & Komodakis, 2016): Wide Residual Network further exploits the "skip connections" used in ResNet and in the meanwhile increases the width of residual networks. In detail, we use the 16 layer Wide ResNet with 4 multipliers (WRN-16-4) in our experiments.

**LSTM** (Hochreiter & Schmidhuber, 1997): Long Short-Term Memory (LSTM) network is a special kind of Recurrent Neural Network (RNN), capable of learning long-term dependencies. It was introduced by Hochreiter & Schmidhuber (1997), and was refined and popularized in many followup work.

### C.3 PARAMETER SETTINGS

We perform grid searches to choose the best hyper-parameters for all algorithms in both image classification and language modeling tasks. For the base learning rate, we do grid search over $\{0.0001, 0.001, 0.01, 0.1, 1, 10, 100\}$ for all algorithms, and choose the partial adaptive parameter $p$ from $\{2/5, 1/4, 1/5, 1/8, 1/16\}$ and the second order moment parameter $\beta_2$ from $\{0.9, 0.99, 0.999\}$. Specifically, in terms of image classification experiments, for SGD with momentum, the base learning rate is set to be $0.1$ with a momentum parameter of $0.9$. For all adaptive gradient methods except Padam, we set the base learning rate as $0.001$. For AdaBound, the final learning rate is set to be $0.1$. For Padam, the base learning rate is set to be $0.1$ and the partially adaptive parameter $p$ is set to be $1/8$ due to its best empirical performance. For Adam, Amsgrad, the momentum parameters are set to be $\beta_1 = 0.9$, $\beta_2 = 0.99$. And for other adaptive gradient methods including Padam, we set $\beta_1 = 0.9$, $\beta_2 = 0.999$. The weight decay factor is set to $5 \times 10^{-4}$ for all methods except AdamW which adopts a different weight decay scheme. For AdamW, the normalized weight decay factor is set to $2.5 \times 10^{-2}$ for CIFAR and $5 \times 10^{-2}$ for ImageNet. For Yogi, $\epsilon$ is set as $10^{-3}$ as suggested in the original paper. For AdaBound, the final learning rate is set as $0.1$ as suggested in Luo et al. (2019). The minibatch sizes for CIFAR-10 and CIFAR-100 are set to be $128$ and for ImageNet dataset we set it to be $256$. Regarding the LSTM experiments, for SGD with momentum, the base learning rate is set to be $1$ for 2-layer LSTM model and $10$ for 3-layer LSTM. The momentum parameter is set to be $0.9$ for both models. For all adaptive gradient methods except Padam and Yogi, we set the base learning rate as $0.001$. For Yogi, we set the base learning rate as $0.01$ for 2-layer LSTM model and $0.1$ for 3-layer LSTM model. For Padam, we set the base learning rate as $0.01$ for 2-layer LSTM model and $1$ for 3-layer LSTM model. For all adaptive gradient methods, we set $\beta_1 = 0.9$, $\beta_2 = 0.999$. In terms of algorithm specific parameters, for Padam, we set the partially adaptive parameter $p$ as $0.4$ for 2-layer LSTM model and $0.2$ for 3-layer LSTM model. For AdaBound, we set the final learning rate as $10$ for 2-layer LSTM model and $100$ for 3-layer LSTM model. For Yogi, $\epsilon$ is set as $10^{-3}$ as suggested in the original paper. The weight decay factor is set to $1.2 \times 10^{-6}$ for all methods except AdamW which adopts a different weight decay scheme. For AdamW, the normalized weight decay factor is set to $4 \times 10^{-4}$. The minibatch size is set to be $20$ for all LSTM experiments.

## D  ADDITIONAL EXPERIMENTS

As a sanity check, we first plot the max and min of $\widehat{\mathbf{v}}_t$ across all the coordinates in Figure 5. In detail, the maximum value of $\widehat{\mathbf{v}}_t$ is around $0.04$ in the end and the minimum value is around $3 \times 10^{-8}$,

$\max\{(\widehat{\mathbf{v}}_t)_i^{1/8}\} - \min\{(\widehat{\mathbf{v}}_t)_i^{1/8}\} \approx 0.55$. We can see that the effective learning rates for different coordinates are quite different.

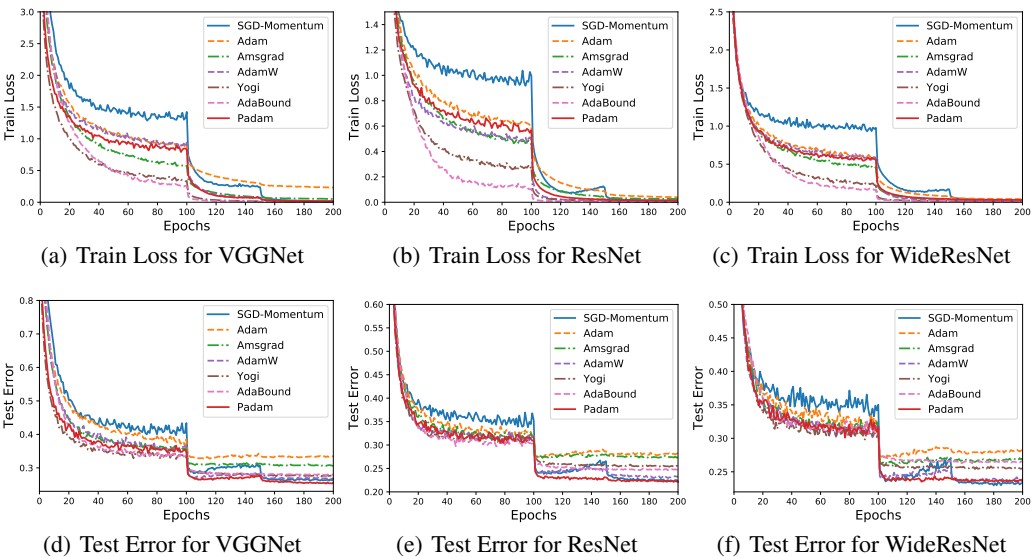

(a) Train Loss for VGGNet      (b) Train Loss for ResNet      (c) Train Loss for WideResNet

(d) Test Error for VGGNet      (e) Test Error for ResNet      (f) Test Error for WideResNet

Figure 6: Train loss and test error (top-1 error) of three CNN architectures on CIFAR-100. In all cases, Padam achieves the fastest training procedure among all methods and generalizes as well as SGD with momentum.

Figure 6 plots the train loss and test error (top-1 error) against training epochs on the CIFAR-100 dataset. We can see that Padam achieves the best from both worlds by maintaining faster convergence rate while also generalizing as well as SGD with momentum in the end.

Tables 1 and 2 show the test accuracy of all algorithms on CIFAR-10 dataset and CIFAR-100 dataset respectively. On CIFAR-10 dataset, methods such as Adam and Amsgrad have the lowest test accuracies. Even though followup works such as AdamW, Yogi, AdaBound improve upon original Adam, they still fall behind or barely match the performance of SGD with momentum. In contrast, Padam achieves the highest test accuracy in VGGNet and WideResNet for CIFAR-10, VGGNet and ResNet for CIFAR-100. On the other two tasks (ResNet for CIFAR-10 and WideResNet for CIFAR-100), Padam is also on a par with SGD with momentum at the final epoch (differences less than $0.2\%$). This suggests that practitioners should once again, use fast to converge partially adaptive gradient methods for training deep neural networks, without worrying about the generalization performances.

Table 1: Final test accuracy of all algorithms on CIFAR-10 dataset. Bold number indicates the best result.

| Models | Test Accuracy (%) | | | | | | |
|---|---|---|---|---|---|---|---|
| | SGDM | Adam | Amsgrad | AdamW | Yogi | AdaBound | Padam |
| VGGNet | 93.71 | 92.21 | 92.54 | 93.54 | 92.94 | 93.28 | **93.78** |
| ResNet | **95.00** | 92.89 | 93.53 | 94.56 | 93.92 | 94.16 | 94.94 |
| WideResNet | 95.26 | 92.27 | 92.91 | 95.08 | 94.23 | 93.85 | **95.34** |

Table 3 shows the final test accuracy of all algorithms on ImageNet dataset. Again, we can observe that Padam achieves the best test accuracy on VGGNet (both Top-1 and Top-5) and Top-1 accuracy on ResNet. It stays very close to the best baseline of Top-1 accuracy on ResNet model.

Table 4 shows the final test perplexity of all algorithms on the Penn Treebank dataset. As we can observe from the table, Padam achieves the best (lowest) test perplexity on both 2-layer LSTM and 3-layer LSTM models.

Table 2: Final test accuracy of all algorithms on CIFAR-100 dataset. Bold number indicates the best result.

| Models | Test Accuracy (%) | | | | | | |
|---|---|---|---|---|---|---|---|
| | SGDM | Adam | Amsgrad | AdamW | Yogi | AdaBound | Padam |
| VGGNet | 73.32 | 66.60 | 69.40 | 73.03 | 72.35 | 72.00 | **74.39** |
| ResNet | 77.77 | 71.72 | 72.62 | 76.69 | 74.55 | 75.29 | **77.85** |
| WideResNet | **76.66** | 71.83 | 73.02 | 76.04 | 74.47 | 73.49 | 76.42 |

Table 3: Final test accuracy on ImageNet dataset

| Models | Test Accuracy (%) | | | | | | |
|---|---|---|---|---|---|---|---|
| | SGDM | Adam | Amsgrad | AdamW | Yogi | AdaBound | Padam |
| ResNet Top-1 | **70.23** | 63.79 | 67.69 | 67.93 | 68.23 | 68.13 | 70.07 |
| ResNet Top-5 | 89.40 | 85.61 | 88.15 | 88.47 | 88.59 | 88.55 | **89.47** |
| VGGNet Top-1 | 73.93 | 69.52 | 69.61 | 69.89 | 71.56 | 70.00 | **74.04** |
| VGGNet Top-5 | 91.82 | 89.12 | 89.19 | 89.35 | 90.25 | 89.27 | **91.93** |

Table 4: Final test perplexity (lower is better) on Penn Treebank dataset

| Models | Test Perplexity | | | | | | |
|---|---|---|---|---|---|---|---|
| | SGDM | Adam | Amsgrad | AdamW | Yogi | AdaBound | Padam |
| 2-layer LSTM | 63.37 | 61.58 | 62.56 | 63.93 | 64.13 | 63.14 | **61.53** |
| 3-layer LSTM | 61.22 | 60.44 | 61.92 | 63.24 | 60.01 | 60.89 | **58.48** |

