# OpenReview forum: "Training Deep Neural Networks with Partially Adaptive Momentum"
_ICLR.cc/2020/Conference — Reject_

### Official Review · AnonReviewer1 · 2019-10-19
**Official Blind Review #1**

**Rating:** 1

**Review:**

% post author response %
Thanks for your detailed response.

R1. Note that in almost all classical optimization routines, the learning rate has a (very intuitive) scaling on the problem parameters - for e.g. in gradient descent, the learning rate looks like 1/smoothness. This is mirrored in the definition of Newton methods - where, in the direction of hessian^{-1} grad, one uses a scale free line search to estimate an appropriate stepsize between 0 and 1 (to emphasize, this value has *no* dependence on problem scaling). While this begins to fall apart with the case of adaptive gradient methods, how can we even hope to justify the potentially arbitrary power of, say, 1/4 or 1/8 used by Padam? This is the reason behind my comment that the algorithm is unnatural. By using such a power of the smoothness of the problem, the other component of the learning rate (alpha_t) is no longer a scale free quantity. It has to depend on other problem dependent parameters for the overall learning rate to be scaled appropriately based on the problem characteristics.

R2. While the paper performs grid search (sec C.3 in the paper) for the partially adaptive parameter, the lambda value for YOGI is set as one suggested in their paper. I can accept the claim of authors if I see more experiments tuning the lambda parameter for YOGI as well.  Otherwise, I dont quite see why one specific parameter value for lambda works for every problem.

R3/R5. My point is that the original paper for adaptive methods (adam/adagrad) never mentions learning rate decay. This seems to have been added in subsequently just to boost the performance. What I do not understand is what is the specific advantage of adaptive methods over SGD if every component used by SGD (including learning rate decay) is used even by adaptive methods. Even from a theoretical bound perspective, to the best of my knowledge, there is no clear indication that the partially adaptive momentum methods actually improve over vanilla sgd (+momentum) in settings that do not involve dense parameters/gradients.

R4. Again, the learning rate decay has a specific use in the papers I referenced in my review. Somehow, the bounds presented in the paper do not reflect the use of a step decay schedule on the learning rates, so, I see that the theorem (aside from the assumptions mentioned) is detached from the practical results even in this respect.
%%

This paper considers generalization issues experienced by adaptive gradient methods compared to well-tuned SGD + momentum, a topic that is of interest in the development of optimization methods for deep learning. The paper is well-written and elaborates on (i) issues faced by adaptive gradient methods in contrast to standard SGD + momentum, (ii) presents experimental results on training standard conv-net based architectures on image classification benchmarks and on training LSTMs on PTB and (iii) presenting theoretical analysis relating convergence of the method to a first-order critical point for smooth stochastic non-convex optimization.

I have questions about certain aspects of the paper, which I will elaborate below:

— If one takes a step back to understand the origins of diagonal adaptation methods (introduced by Adagrad), this was motivated by the infeasibility of using the inverse square root of a full pre-conditioning adaptation matrix. If we think of such full matrix adaptation methods, is this paper implying the use of other matrix powers (other than a square root) as used by adagrad (or other adaptation approaches)? This appears very unnatural to me.

-- If the main issue is that it is not possible to typically use a large learning rate at the start with ADAM or other preconditioning methods and that leads to using other powers of the diagonal adaptation matrix, a more natural fix would be to use a conservative (trust region inspired) approach to reduce aggressive steps at the start. What I mean is as follows: Suppose H_t is a preconditioning matrix, g_t is the gradient (or the discounted sum of gradients with/without bias correction). The current approach is to make H_t diagonal and use H_t ^{-1/2}g_t as the update matrix. One option to prevent aggressive initial steps is to use (H_t + \lambda)^{-1/2} g_t, either with having lambda fixed through the optimization or making it a function of iterations.

— I am not aware that papers which employ adaptive methods also tend to use some form of learning rate decay (on the alpha_t ’s) - at least, by looking at the original papers of Adam/adagrad, I do not see the combination of learning rate decay and adaptive methods. In a sense, if one has an `"adaptive" optimization method, it’d be unnatural to have to use some form of step decay of the learning rates (alpha_t's) in conjunction with these methods. One would typically just use SGD+momentum with some form of such a step decay of the learning rates. This to me is a serious shortcoming - it appears to make the use of adaptive methods almost irrelevant because the only hyper-parameter that it gets rid of is the dependence on the initial learning rate (because, for SGD, we anyway have the other hyper-parameters like momentum, stepdecay factor, when to decay learning rate etc.)

— With regards to theory (and connections to experiments): Despite the fact that the paper employs a step decay schedule on the learning rates (alpha_t), their theorem statement (or any corollary) doesn’t actually employ this specific step size decay scheme (on the alpha_t ’s) and attempts to understand what are the advantages of the step decay schedule on the convergence statements provided.  The step decay schedule has featured in several recent efforts in the stochastic optimization community (both with convex (https://arxiv.org/pdf/1607.01027.pdf, https://arxiv.org/pdf/1904.12838.pdf)/non-convex (e.g. https://arxiv.org/pdf/1907.09547.pdf) objectives), where, the results indicate non-trivial advantages of using these step-decay schemes on alpha_t’s (though, these are with non-adaptive optimization methods).

— Along these lines (of the previous point), it is important to note what the performance of the optimization method is when alpha_t’s are fixed to a specific value (without being decayed over the course of optimization) - since this relates to the most standard definition (and advantage) associated with using adaptive gradient methods. My understanding is that this result continues to be fairly sub-optimal compared to using SGD+momentum with a step decay schedule.

Other minor comments:
— I do not understand the use of the term “second order” momentum for calling variables that have a running average of squared gradients. This term is misleading in what it represents.


**Experience Assessment:**

I have published one or two papers in this area.

**Review Assessment: Checking Correctness Of Derivations And Theory:**

I assessed the sensibility of the derivations and theory.

**Review Assessment: Checking Correctness Of Experiments:**

I assessed the sensibility of the experiments.

**Review Assessment: Thoroughness In Paper Reading:**

I read the paper at least twice and used my best judgement in assessing the paper.

---

> ### Author Response · Authors · 2019-11-13
> **Response to Reviewer #1**
>
> Thank you for your valuable comments. We address your questions as follows.
>
> Q1: “If one takes a step back to  ...appears very unnatural to me.”
>
> R1. As we mentioned in the introduction part, using the standard adaptation matrix could lead to poor generalization performances in practice due to the “small learning rate dilemma”. We control the adaptiveness by introducing a partial adaptiveness parameter $p$, which interpolates between SGD with momentum ($p=0$)  and Adam $p=1/2$. We believe this is quite intuitive and is a natural way to do it.
>
> Q2: “If the main issue is that it is not possible …  the optimization or making it a function of iterations.
>
> R2. Thanks for suggesting a solution to this problem. However, what you present about adding a $\lambda$ term on $H_t$ has already been proposed in the algorithm Yogi (Zaheer et al., NuerIPS’2018), which we already commented and compared with as one of the baseline methods. Our empirical evaluations have shown that Padam can achieve better generalization performance over your proposed solution.
>
> Q3: “I am not aware that papers which … step decay factor, when to decay learning rate etc.)”
>
> R3. We think you might have a misunderstanding towards adaptive gradient methods here. We would like to clarify that there is no contradiction between learning rate decay and adaptive gradient methods, as adaptive here only refers to different learning rate for different coordinates, and learning rate decay is applied upon the base learning rate, which is complementary to adaptiveness in different coordinates. As you can see from Theorem 4.3, we choose $\alpha_t$ as a constant $\alpha$, and the learning rate decay schedule only applied on this $\alpha$ while adaptiveness is applied on $\hat v_t$ which controls the different levels of learning rate for different coordinates.
> In practice, learning rate decay has been shown effective for most of the algorithms. For example, a lot of the recent literatures have already used the various forms of learning rate decay schedule to achieve the state-of-the-art performances, for example, (Luo et al., 2019) (Loshchilov & Hutter, 2019). In fact, without the learning rate decay schedule, adaptive gradient method cannot achieve comparable/better results than SGD with learning rate decay.
>
> Q4: “With regards to theory ...advantages of using these step-decay schemes on alpha_t’s”
>
> R4. Thank you for pointing out the papers. We have commented on these works in the revision. Given that our main focus of this paper is not about studying the effects of learning rate decay (as can be seen in the suggested papers, studying the effects of learning rate decay alone is already highly non-trivial and can be an independent topic), but to improve adaptive gradient methods to better train deep neural networks, we simply choose the current best learning rate decay schedule that gives the highest performance boost for all methods.
>
> We would also like to point out that in Xu et al., 2016 and Davis et al., 2019, a key reason for the appealing theoretical guarantee is that at each stage, the input is chosen as the average of iterates in the previous stage, which is different from common practice of training deep neural networks. In addition, Ge et al., 2019 only studies convex quadratic objective functions. Therefore, there is still a huge gap between these theoretical results and the practice on learning rate decay schedules, and applying the analyses in these works seems not sufficient to obtain the desired theoretical guarantees for Padam with stagewise learning rate decay.
>
> Q5: “it is important to note...compared to using SGD+momentum with a step decay schedule.”
>
> R5. We respectfully disagree with your opinion about “performance of the optimization method”. What you suggested is not a fair comparison. If what you want is to compare the performances without further stage-wise learning rate decay schedule, we should remove it for all methods (as this decaying schedule is not tied to or conflict with any specific optimization method).
>
> With our focus on better training deep neural networks to achieve the state-of-the-art empirical generalization performance, we believe that the fair way is to compare different methods using their best achievable performances. Empirical findings suggest that applying the learning rate decay schedule is always better than not applying it, for all the methods we compared. Therefore we adopt this learning rate schedule for all methods.
>
> Q6: “the term “second order” momentum ...in what it represents.”
>
> R6. We would like to clarify that this is a typo. We meant to use the term “second order moment”, which  is originated from the original Adam paper, where it says “estimates of first and second moments of the gradients”. In fact, the name Adam stems from adaptive (first and second) moment estimation. We have fixed these typos in the revision.

---

### Official Review · AnonReviewer3 · 2019-10-25
**Official Blind Review #3**

**Rating:** 6

**Review:**

It has been empirically observed that adaptive optimizers such as Adam/Amsgrad
lead to worse generalization than SGD + momentum when used to train neural
networks.
Motivated by this observation the authors suggest Padam, a modification of Adam/Amsgrad.
Padam contains a parameter p that, when set to 0 reduces their method to SGD + momentum
and when set to 1/2 reduces their method to Adam/Amsgrad.

The Padam iteration is roughly $x^{k+1} = x^k - \alpha * g / v^p$
where $g$ is an exponential moving average of stochastic gradients and
$v$ is an exponential moving average of the squared gradient.

In that way Padam is capable of interpolating between the two methods in order
to find a good trade-off between the improved convergence of the one and the
improved generalization capability of the other.

The authors also suggest an explanation for why the generalization gap
of Adam happens:
They claim that it is due to the "small-learning rate dilemma" that happens
as follows.
A small second moment of the stochastic gradients as approximated by $v^{1/2}$
(think variance) in can lead to large effective steps in some components.
To balance this effect out, Adam needs to choose smaller step sizes than
SGD + momentum.
If the same learning rate schedule is used based on a smaller base step size,
Adam under-trains at the end of training.

The authors suggest that Padam with p < 1/2 can use larger learning rates
because it does not have as large effective steps if second moments are small.

The authors also prove convergence rates for making gradients small with Padam.
They use the setting of nonconvex optimization and not online convex regret
analysis like the Adam/Amsgrad papers.

Finally, the presented experiments using image data sets and standard neural
network architectures suggest that Padam indeed shares the advantages of both
SGD + momentum and Adam and thus obtains a best of both worlds.

I suggest to accept the paper.
It introduces an elegant generalization of Amsgrad that appears to be
empirically useful in experiments.
The experiments seem to be fairly performed (in terms of hyperparam search).
The rigorous convergence analysis is laudable although perhaps not as relevant
as the practical usefulness of the suggested approach.

I do however suggest that some changes be made.
1. The "small learning rate dilemma" phenomenon needs to be more clearly defined
	and explained.
	The whole relationship of SGD step size schedules to generalization
	(e.g. simulated annealing analogy) is certainly nontrivial.
	I would rather the paper not make some nonrigorous claims if there is no
	proof.
	Or state clearly whether something is conjecture or proposition (with proof).

2. Explain why the same learning rate schedule should or is (in practice) used
	for Adam as for SGD + momentum, as this does not seem to make sense given
	the aforementioned dilemma.

Specific notes / suggestions:
- Page 2: "We proposed a novel" -> "We propose a novel"

- Page 4: "bridging this generalization gap"
	"this" does not make sense to me in that context, rather use "the"

- Page 5: Figure 1, p = 1/16 seems to still be the most attractive in terms of
	generalization in the long run. Why would I not want to wait till the model
	is fully trained?

- Page 4: "It is very likely that Adam/Amsgrad is "over-adaptive"
	This seems to me a strong claim and the explanation that follows it to me
	is not rigorous enough.


**Experience Assessment:**

I have read many papers in this area.

**Review Assessment: Checking Correctness Of Derivations And Theory:**

I assessed the sensibility of the derivations and theory.

**Review Assessment: Checking Correctness Of Experiments:**

I assessed the sensibility of the experiments.

**Review Assessment: Thoroughness In Paper Reading:**

I read the paper at least twice and used my best judgement in assessing the paper.

---

> ### Author Response · Authors · 2019-11-13
> **Response to Reviewer #3**
>
> Thank you for your supportive comments. We address your concerns as follows.
>
> Q1: “The "small learning rate dilemma" phenomenon ... is conjecture or proposition (with proof)”
>
> R1. Thank you for your suggestion. The “small learning rate dilemma” phenomenon is an empirical observation. We have better described and further explained the “small learning rate dilemma” in Section 3 in the revision.
>
> Q2: “Explain why the same learning rate schedule... make sense given the aforementioned dilemma.”
>
> R2. This is simply because in practice, stage-wise learning rate decay gives better generalization performances. If we do not apply the stage-wise learning rate decay strategy, it will end up with worse generalization performance, as can be seen in our Figure 2 (d)(e)(f), the test error is barely decreasing around 100th epoch. Thus we apply the stage-wise learning rate decay strategy for all methods for a fair comparison. On the other hand, there is no contradiction between this learning rate decay schedule and the “small learning rate dilemma”.
> The “small learning rate dilemma” does not deny the effect of learning rate decay on adaptive gradient methods, and only suggests that the effect of stage-wise learning rate decay on adaptive gradient methods is not as significant as that on SGD + momentum.  We have clarified this in Section 3 in the revision.
>
> Q3: “Figure 1, $p = 1/16$ seems to still... Why would I not want to wait till the model is fully trained?
>
> R3. Figure 1 is an illustration of different choice of $p$, we have updated the plot with a fully trained one. In experiments, we do grid search to choose the best $p$.
>
> Q4: "It is very likely that Adam/Amsgrad is "over-adaptive" This seems to me a strong claim and the explanation that follows it to me is not rigorous enough”
>
> R4. Thank you for your suggestion. We have rephrased this claim and provided more explanation in the revised version.

---

### Official Review · AnonReviewer2 · 2019-10-26
**Official Blind Review #2**

**Rating:** 3

**Review:**

This paper proposes a new variation on adaptive learning rate algorithm that unifies SGD with momentum and Adam/Amsgrad. They provided convergence proof for this algorithm. The effectiveness of the proposed method was demonstrated through various domains and neural networks architecture. Though the empirical results are extensive, I am leaning towards reject because (1) The reason why the method works isn't clear. (2) The theory doesn't justify the practice. (3) The practical usefulness of the algorithm isn't clear. Here are my detailed comments:

(1) The paper provides an observation which they call "small learning rate dilema": One often uses a smaller base learning rate for adaptive gradient methods than SGD with momentum. This makes the boost one can gain from applying learning rate decay to adaptive gradient methods not as significant as applying to SGD with momentum. Based on this observation, they propose to penalize the adaptiveness by adjusting the value p in their algorithm. However, the proposed adjustment seems like a trivial one, without giving too much insights into why learning rate decay is not compatible to adaptiveness. An insightful analysis should try to first answer the following questions:
      (a) Why shall one start with a large learning rate in the beginning?
      (b) Why does learning rate decay gives a boost to performance?
      (c) If one uses an adaptive method, how does it affect one's choice for the initial learning rate?
      (d) and how does the adaptiveness changes the effect of learning rate decay?
To answer those questions, I suggest the author to read [1] where they gave partial answers to (a) (b). If one tries to do similar analysis performed in [1] for adaptive methods, one might be able to answer (c) and (d).

(2) I have two criticisms to the theoretical analysis carried out in the paper. The most important issue is that the analysis is not useful to show the effectiveness of the proposed method:
     (a) The rate of convergence matches with SGD + momentum. So it is not better than the baseline.
     (b) It does not show the relationship of adaptiveness and decaying learning rate schedule.
The second criticism is related to the novelty of the theorems.  Please correct me on this because I did not go over the theorems carefully. But based on my crude assessment,  theorems are mostly mechanical applications of prior work to the current extended version. Hence it does not provide any further insights into the convergence of nonconvex optimization methods.

(3) Though the empirical results are good, where the proposed method matched or outperformed all previous methods in The method introduces one extra hyperparameter, p, for tuning. It is then questionable whether the algorithm is efficient in terms of hyperparameter searches, i.e., how many hyparameter sweeps are needed for finding a good run, versus baseline methods like SGD+momentum. Since the performance of the methods are mostly the same, if the proposed method requires as many hyperparameter tuning as SGD+momentum, then it makes the proposed method less useful in practice.

[1] Understanding short-horizon bias in stochastic meta-optimization. Wu et. al. ICLR 2018.

**Experience Assessment:**

I have published in this field for several years.

**Review Assessment: Checking Correctness Of Derivations And Theory:**

I assessed the sensibility of the derivations and theory.

**Review Assessment: Checking Correctness Of Experiments:**

I carefully checked the experiments.

**Review Assessment: Thoroughness In Paper Reading:**

I read the paper at least twice and used my best judgement in assessing the paper.

---

> ### Author Response · Authors · 2019-11-13
> **Response to Reviewer #2**
>
> Thank you for your constructive comments. We address your questions as follows.
>
> Q1: “However, the proposed adjustment seems like...one might be able to answer (c) and (d)”
>
> R1. We want to clarify the relationship between learning rate decay and the new adaptive method we proposed. Recall that the effective learning rate in our paper is defined as $\alpha / {\hat v_t}^p$, where $\alpha$ is the universal learning rate and $\hat v_t$ is the normalization term for the adaptive gradient.  The learning rate decay schedule only applies to this $\alpha$ while the partial adaptiveness is controlled by ${\hat v_t}^p$. Note that the main focus of this paper is not about finding the best learning rate decay schedule regarding $\alpha$, but designing a new algorithm to control the adaptiveness for better empirical generalization result.
>
> For this reason, we do not directly consider learning rate decay in our theoretical analysis (which is also discussed in R2b). On the other hand, we do implement learning rate decay in our experimental results following our intuition. This also demonstrates that Padam can achieve state-of-the-art performance.
>
> We agree that a thorough analysis on how learning rate decay affects adaptive gradient methods is an important direction, which we have discussed in the future work section.  Also thank you for pointing out the nice paper by Wu et al. 2018. Since their analysis is for simple quadratic cost functions, extending their analysis to general nonconvex functions studied in this paper is nontrivial, and we will explore it in our future work.
>
> Q2a: “The rate of convergence matches with SGD + momentum. So it is not better than the baseline”
>
> R2a. Regrading the convergence rate, to the best of our knowledge, in nonconvex setting, all existing adaptive gradient methods can only achieve the same rate as SGD using current proof technique for adaptive gradient methods. Showing theoretically the advantage of adaptive gradient methods over SGD is still an open problem. Although we cannot prove a better convergence result than SGD, we showed in Remark 4.6 that the convergence rate of Padam is indeed better than that of Adam/Amsgrad.
>
> Q2b: “It does not show the relationship of adaptiveness and decaying learning rate... optimization methods”
>
> R2b: It is true that our current results does not show the relationship of adaptiveness and decaying learning rate schedule. However, we would like to emphasize that for adaptive gradient methods, especially our newly proposed Padam algorithm, even the more standard setting without learning rate decay has not been well-studied. Skipping this more standard setting and directly studying Padam with learning rate decay may only give unclear and jumbled results. Therefore in this paper we choose to focus on fixed learning rate in the convergence analysis. Only after this type of theoretical results has been established, can the theoretical analysis of combining Padam/Adam and learning rate decay be meaningful and insightful. We consider such a combination as an important future work direction, and  have added discussions in the future work section.
>
> Q3 “Though the empirical results are good, ... then it makes the proposed method less useful in practice”
>
> R3. Our proposed method indeed needs one more parameter to unify SGD and Adam. That said, we argue that  tuning this extra parameter is actually easy: just like Adam with learning rate 0.001 works for many cases, we found that $p = 1 / 8$ is generally good for various tasks and architectures. Therefore, it usually takes only a few trials to find the best p in practice. In fact, to achieve better empirical performances, newly proposed methods such as AdaBound (Luo et al., 2019), Yogi (Zaheer et al., 2018), AdamW (Loshchilov & Hutter, 2019) all requires tuning at least one more hyper-parameter.

---

### Decision · Program_Chairs · 2019-12-19

**Decision:**

Reject

**Comment:**

This paper extends Adam by adding another hyperparameter that allows the second moments to be raised to a power p other than 1/2. This certainly seems worth trying. The paper is well written, and the experiments seem reasonably complete. But some of the reviewers and I feel like the contribution is a bit obvious and incremental. The "small learning rate dilemma" needs a bit more justification: since the denominator has a different scale, the learning rates for different values of p are not directly comparable. It could very well be that Adam's learning rate has to be set too small due to some outlier dimensions, but showing this would require some evidence. From the experiments, it does seem like there's some practical benefit, though it's not terribly surprising that adding an additional hyperparameter will result in improved performance. The reviewers think the theoretical analysis is a straightforward extension of prior work (though I haven't checked myself). Overall, it doesn't seem to me like the contribution is quite enough for publication at ICLR.